# Immunological history governs human stem cell memory CD4 heterogeneity via the Wnt signaling pathway

Hassen Kared [1]*, Shu Wen Tan[1], Mai Chan Lau[1], Marion Chevrier [1], Crystal Tan[1], Wilson How[1], Glenn Wong[1], Marie Strickland [1,2], Benoit Malleret [1,3], Amanda Amoah[4], Karolina Pilipow[5], Veronica Zanon[5], Naomi Mc Govern[1], Josephine Lum[1], Jin Miao Chen[1], Bernett Lee[1], Maria Carolina Florian[4], Hartmut Geiger[4,6], Florent Ginhoux [1], Ezequiel Ruiz-Mateos[7], Tamas Fulop[8], Reena Rajasuriar[9,10,11], Adeeba Kamarulzaman[9,11], Tze Pin Ng[12], Enrico Lugli [5] & Anis Larbi[1,3,8]*

The diversity of the naïve T cell repertoire drives the replenishment potential and capacity of memory T cells to respond to immune challenges. Attrition of the immune system is associated with an increased prevalence of pathologies in aged individuals, but whether stem cell memory T lymphocytes ($T_{SCM}$) contribute to such attrition is still unclear. Using single cells RNA sequencing and high-dimensional flow cytometry, we demonstrate that $T_{SCM}$ heterogeneity results from differential engagement of Wnt signaling. In humans, aging is associated with the coupled loss of Wnt/β-catenin signature in CD4 $T_{SCM}$ and systemic increase in the levels of Dickkopf-related protein 1, a natural inhibitor of the Wnt/β-catenin pathway. Functional assays support recent thymic emigrants as the precursors of CD4 $T_{SCM}$. Our data thus hint that reversing $T_{SCM}$ defects by metabolic targeting of the Wnt/β-catenin pathway may be a viable approach to restore and preserve immune homeostasis in the context of immunological history.

[1] Singapore Immunology Network (SIgN), Agency for Science Technology and Research (A*STAR), Immunos Building, 8A Biomedical Grove, Biopolis, Republic of Singapore. [2] Clinical and Experimental Sciences, Faculty of Medicine, University of Southampton, Southampton, UK. [3] Department of Microbiology and Immunology, Yong Loo Lin School of Medicine, National University of Singapore, Singapore, Republic of Singapore. [4] Institute of Molecular Medicine, University of Ulm, Ulm, Germany. [5] Humanitas Clinical and Research Center, Laboratory of Translational Immunology (LTI), Rozzano, Italy. [6] Experimental Hematology and Cancer Biology, CCHMC, Cincinnati, OH, USA. [7] Clinical Unit of Infectious Diseases, Microbiology and Preventive Medicine, Institute of Biomedicine of Seville (IBiS), Virgen del Rocío University Hospital, CSIC, University of Seville, Seville, Spain. [8] Department of Medicine, Faculty of Medicine, University of Sherbrooke, Sherbrooke, Quebec, Canada. [9] Centre of Excellence for Research in AIDS (CERiA), University of Malaya, Kuala Lumpur, Malaysia. [10] The Peter Doherty Institute for Infection and Immunity, University of Melbourne, Melbourne, Victoria, Australia. [11] Faculty of Medicine, University of Malaya, Kuala Lumpur, Malaysia. [12] Gerontology Research Programme and Department of Psychological Medicine, Yong Loo Lin School of Medicine, National University of Singapore, Singapore, Singapore. *email: hassen.acn@gmail.com; anis_larbi@immunol.a-star.edu.sg

The stem-cell-like self-renewal ability is a vital feature of immune memory T cells that preserves the lifetime health of an individual[1,2]. Naturally acquired immunity or the induction of long-term protection occurs during primary infection and vaccination[3,4]—even in the absence of re-exposure to the respective immunizing agent. The maintenance of immunological memory requires the preservation of T-cell stemness and the flexibility of various T-cell compartments to alternate between a persistent quiescent state and a proliferative state where specific clones divide asymmetrically to give rise to activated memory and effector T cells[5,6], which are vital for pathogen clearance during antigen re-exposure[7].

The recent discovery of stem-cell memory T lymphocytes ($T_{SCM}$) fills an important void that had hitherto obscured our understanding of the ontogeny of memory T cells. $T_{SCM}$ represent a discrete but phenotypically detectable population in animal models (mice, nonhuman primates) and humans[8–10]. These cells have the ability not only to self-renew but also to differentiate into all subsets of memory and effector T cells[11]. Combined with their longevity, the preservation of $T_{SCM}$ plasticity may play a central role in maintaining immunologic competence with age[12]. Their essential role in the development of autoimmune disorders, chronic infections, and cancer have already identified them as putative targets for future vaccines or adaptive T-cell therapies[13]. Although the paucity of $T_{SCM}$ in peripheral blood limits the possibility of ex vivo studies, $T_{SCM}$ can evolve from naive T cells following in vitro stimulation with IL-7/IL-15[14], activation of Wnt/β-catenin[9,15] and Notch pathways[16], or modulation of mTOR signaling[17]. Most studies on $T_{SCM}$ have focused on CD8 T cells, but the generation and programming of naive CD4 T cells into $T_{SCM}$ may occur through similar mechanisms. Whether the latter is achieved through specific transcriptional programming of putative $T_{SCM}$-precursors, i.e., recent thymic emigrants (RTE), naive CD4 T cells with homeostatic proliferative history (CD31$^-$), CD103$^+$ naive T cells, memory T cells with a naive phenotype ($T_{MNP}$)[18], or virtual memory T cells[19] remains unknown[20–22]. Moreover, it is unclear whether $T_{SCM}$ are preserved during the process of aging, where the processes of thymic involution[23] and memory compartment inflation become prominent. Age-dependent thymic involution restricts the output of recent thymic emigrants ($T_{RTE}$) that highly express PTK7[24] and CD31[25–27]. This process of immune aging summarizes a collection of immune defects that accumulates over the course of aging, resulting not only from compartmental changes in the representation of immune cells—such as the gradual decline of naive[28] and accumulation of memory differentiated T cells—but also loss of function through immune signaling defects that evolve from intrinsic cellular remodeling.

In this paper, we test the hypothesis that human aging—either chronological or "inflammation induced by chronic HIV infection"—affects $T_{SCM}$ renewal capacity. A combination of flow cytometry phenotyping, single-cell RNA sequencing, confocal imaging, and functional assays supports our inquiry. Our results demonstrate (i) a reduction in $T_{SCM}$ frequencies with age and chronic inflammation; (ii) aging compromises the Wnt/β-catenin signature in CD4 $T_{SCM}$; (iii) inflammation and aging promotes the production of DKK-1 (a natural inhibitor of the Wnt/β-catenin pathway); and (iv) CD4 RTE are the most likely source of peripheral CD4 $T_{SCM}$ cells. Collectively, our data thus reveal a potential for the rejuvenation of the CD4 T-cell compartment through therapeutic targeting of Wnt/β-catenin pathways. Specifically, we may restore loss of $T_{SCM}$ function and diversity that is impacted by immunological history through the calibrated use of Wnt/β-catenin agonists.

## Results

**Depletion of $T_{SCM}$ CD4 cells during aging.** Despite an abundance of literature on the differentiation of CD4 T cells, the ontogeny of naive or early-stage memory CD4 T-cell subsets is poorly understood. Studies generally fail to appreciate their heterogeneity by grouping CD45RO$^-$CCR7$^+$CD27$^+$CD62L$^+$ CD4 T cells into a homogeneous $T_{NAIVE}$ cell compartment, despite their diverse expression of other functional T-cell markers (Supplementary Table 1). We hypothesize that compared with this global population of $T_{NAIVE}$ cells (CD45RO$^-$CCR7$^+$), $T_{SCM}$, given their plasticity, are likely to be more heterogeneous and better sustained in older individuals to compensate for their reduced thymopoiesis. To illustrate this, we characterized T cells within the broad naive phenotype (Fig. 1; Supplementary Fig. 1A) into distinct populations using a combination of high-dimensional flow cytometry, molecular, and single-cell analysis with several analytical tools (including t-SNE, uMAP, Seurat, and diffusion map).

First, CD4 $T_{SCM}$ frequencies demonstrated an even more pronounced age-associated trend than observed for $T_{NAIVE}$ cells ($p < 0.0001$, $n = 43$ and $n = 166$ for young and older donors, respectively, Fig. 2a), the latter may be linked to thymic atrophy as shown by the peripheral decrease of $T_{RTE}$ during aging (Supplementary Fig. 1B, C); we observed a similar trend for CD8 T cells ($p < 0.0001$, Supplementary Fig. 1D). Although both $T_{SCM}$ and $T_{NAIVE}$ frequencies were reduced, a correlation between the two population existed only in older individuals (Fig. 2b, $n = 78$, $r = 0.7188$, $p < 0.0001$), suggesting dysregulated homeostasis during aging.

A leading hypothesis is that enhanced inflammation and chronic infections such as HSV, CMV, dengue, or *Helicobacter pylori* during aging would affect immune homeostasis and contribute to pathology (Supplementary Table 2). Persistent stimulation of virus-specific $T_{SCM}$ CD4 cells might skew their differentiation toward an inflammatory-like state. Levels of pro-inflammatory molecules (Fig. 2c) are significantly elevated in older adults, which aligns with the concept of inflammaging; these elevations are also observed during HIV infection. We, respectively, demonstrate lower absolute CD31$^+$ naive (including $T_{RTE}$ and $T_{SCM}$) and $T_{SCM}$ CD4 T-cell counts in an independent aging ($n = 98$) and HIV-infected cohort ($n = 16$) (Fig. 2d; Supplementary Fig. 1E). This role of HIV in driving inflammation and CD4 depletion is supported by a reversal in the levels of systemic inflammation markers (Galectin-9, sCD163) and CD4 T-cell counts (and subsets)[29] after HAART (Fig. 2e; Supplementary Fig. 1F). Although CD4 $T_{SCM}$ and $T_{CM}$ appeared most susceptible to HIV infection[30], their recoveries were also most pronounced ($p = 0.0004$ and $p < 0.0001$, respectively; $n = 14$). Conversely, the frequencies of late-differentiated $T_{EM}$ was reduced ($p < 0.0001$, $n = 14$) by therapy. Overall, these results are consistent with the hypothesis that lifetime immunological history and levels of inflammation could alter the distribution of CD4 T-cell subsets, interfere with thymus activity[31], and showed reversibility with the allieviation of inflammatory levels.

Next, we examined $T_{SCM}$ heterogeneity via high-dimensional flow cytometry. Here, several clusters of $T_{SCM}$ (from the pool of CCR7$^+$CD45RA$^+$CD27$^+$CD95$^+$) were identified in young and old donors (Fig. 2f, black/red plots, respectively). These clusters exhibited the differential expression of PTK7, CD31, CD127, CD150, and CXCR4. We observed that the representation of individual $T_{SCM}$ clusters was altered during aging. For instance, the population of $T_{SCM}$ co-expressing PTK7 and CD31 (RTE-like) was reduced in older donors (Fig. 2g, $p < 0.05$, $n = 9$ and $n = 13$ for young and elderly donors, respectively). The loss of $T_{NAIVE}$, $T_{RTE}$, and $T_{SCM}$ was exacerbated in older HIV patients,

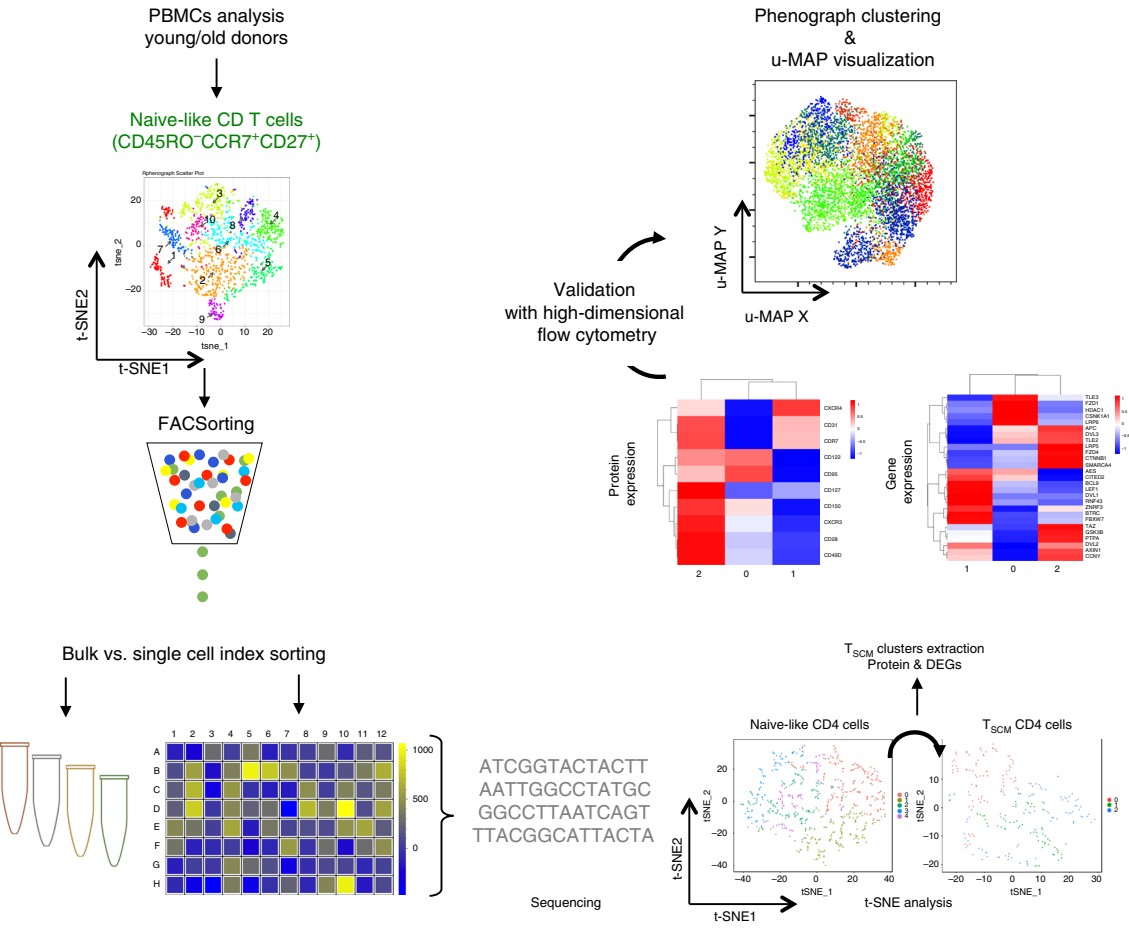

**Fig. 1 Study design.** Workflow of high-dimensional analysis to characterize the heterogeneity of human CD4 $T_{SCM}$ cells. Arrows represent in vitro or in silico experiments, respectively. CCR7$^+$CD45RO$^-$CD27$^+$ CD4 T cells from young ($n = 4$) and elderly donors ($n = 4$) were index sorted and distributed as single cells in 96-well plates (two plates per donor) (1). The MFI corresponding to surface protein expression for each individual cell was recorded for CD28, CD31, CD49d, CD95, CD122, CD127, CD150, CXCR3, CXCR4, and CCR7. A library of DNA sequences was established after the RNA extraction of each individual cell (2). The analysis of single-cell RNA-seq data was reported with t-SNE to identify the clusters of cells with similar profile of gene expression (3). Levels of CD95 protein expression were overlaid on the different clusters to identify clusters that were enriched in $T_{SCM}$ (4). The clusters corresponding to CD4 $T_{SCM}$ were extracted and re-analyzed by t-SNE (5). The heterogeneity of CD4 $T_{SCM}$ was validated with FACS Symphony high-dimensional flow cytometry staining (6).

who also manifested the simultaneous enrichment of $T_{MNP}$ and $T_{EM}$ populations (Supplementary Fig. 1E). Finally, while HAART assists in the transient restoration of $T_{SCM}$ subsets, the absolute counts of $T_{SCM}$ did not normalize even after long-term therapy (Fig. 2d, e).

**Heterogeneity of the $T_{SCM}$ and Wnt signaling pathway.** In addition to the alteration of the above-mentioned markers during aging, we investigated the kinetics of additional markers using an unsupervised approach. Here, "naive" CD4 T cells (CD45RO$^-$ CCR7$^+$CD27$^+$CD62L$^+$) were segregated into distinct populations based on the combinatorial expression of several differentiation-linked markers (Fig. 3a, b). Among ten clusters obtained from flow cytometry results, we were able to identify a cluster with a $T_{SCM}$ phenotype (CCR7$^+$CD45RO$^-$CD27$^+$CD62L$^+$CD122$^+$CD95$^+$, Cluster 5) that could be distinguished from immature $T_{RTE}$ (CCR7$^+$ CD45RO$^-$CD27$^+$CD62L$^+$CD122$^-$CD95$^-$CD31$^{high}$, Cluster 3), $T_{NAIVE}$ (CCR7$^+$CD45RO$^-$CD27$^+$CD62L$^+$CD122$^-$CD95$^-$CD31$^-$, Cluster 10) and $T_{MNP}$ (CCR7$^+$CD45RO$^-$CD27$^+$CD62L$^+$CD122$^-$ CD95$^-$CD31$^-$CD49d$^{high}$CXCR3$^+$, Cluster 8) cells. In order to demonstrate that these subsets were unique at the transcriptomic level, we performed RNA sequencing, including after single-cell index sorting. Regardless of age, CD4 $T_{SCM}$ appeared more

genetically heterogeneous than other CD4 T-cell subsets (Fig. 3c, left panel and Supplementary Fig. 2A, B). $T_{CD31}^{neg}{}_{NAIVE}$, $T_{RTE}$, $T_{MNP}$, and $T_{SCM}$ CD4 T cells exhibited distinctive gene expression profiles (Fig. 3c, right panel), while non-RTE naive and $T_{MNP}$ clustered closely; the ontogenic proximity of the latter subsets can be corroborated by their coordinated modulation of CD28, CD122, CD150, and CXCR3 expression (Supplementary Figs. 1A, 2C, D).

Based on this established $T_{SCM}$ phenotype (based on protein expression measured by flow cytometry during index sorting), we investigated whether genes associated with $T_{SCM}$ generation—specifically the canonical Wnt/β-catenin pathway (Fig. 3d)—were differentially expressed by different $T_{SCM}$ subsets and whether these patterns of expression were altered during aging. First, we observed that $T_{SCM}$ from the young separated into three clusters based on the expression of molecules associated with canonical Wnt/β-catenin signaling (Fig. 3d); this pattern of gene expression is also drastically altered during aging (Fig. 3d). In the gene set enrichment analysis (GSEA, Fig. 3e; Supplementary Fig. 3A) of $T_{SCM}$ transcripts from young donors ($n = 4$), we identified the enrichment of genes that overlapped with noncanonical Wnt–calcium and planar cell polarity (PCP) pathways (Supplementary Fig. 3B) in Clusters 0 and 1, while the expression of RNA transcripts within Cluster 2 overlapped most with canonical

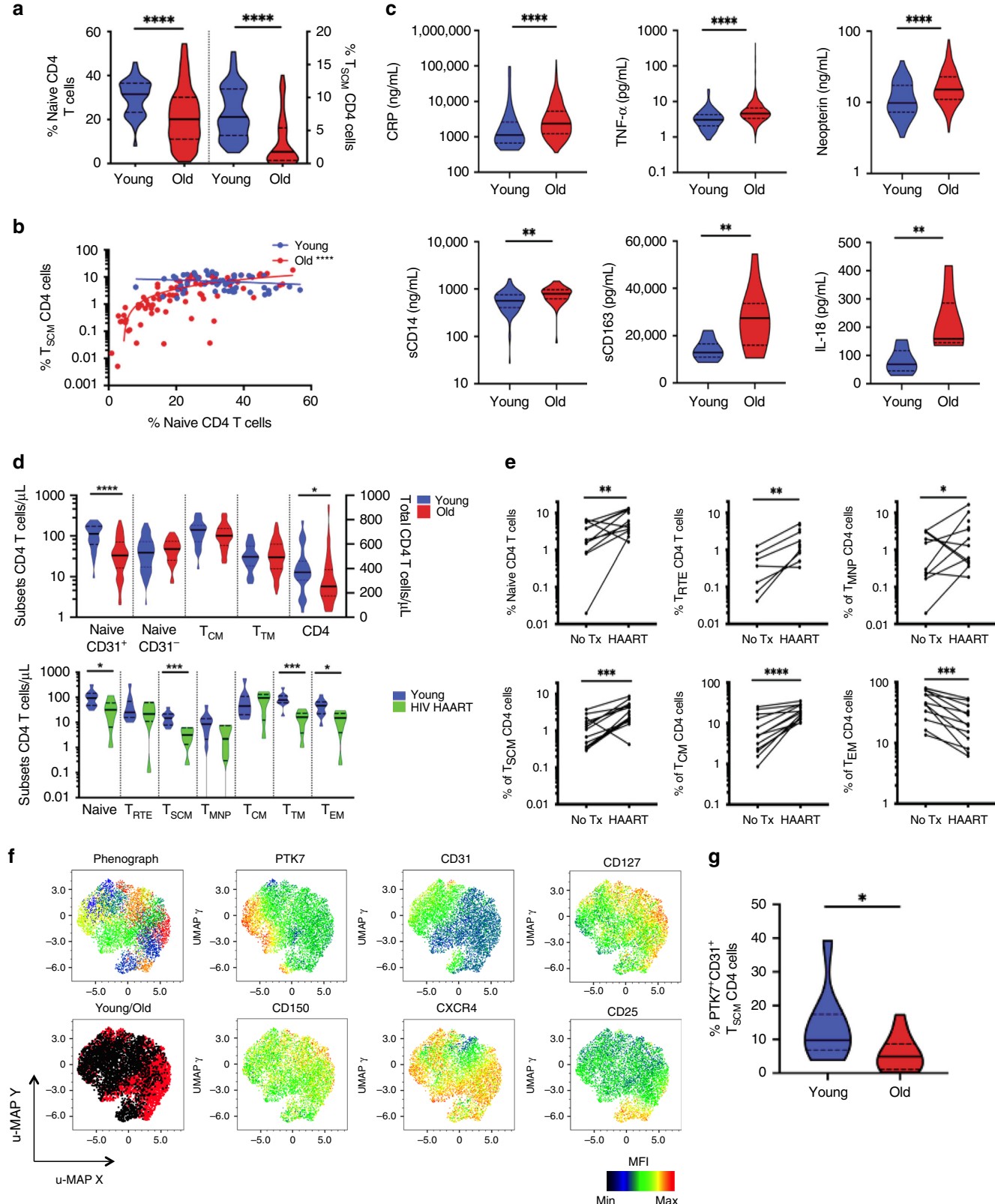

signaling (high *TCF7, LRP5, GSK3B, CTNNB1,* and *FZD4* expression). The specific expression of *NFATC1* and *NLK* or of *RAC1* coupled with *CDC42* and *FZD6* suggested an increased engagement of the calcium and PCP pathway in Clusters 0 and 1, respectively. In addition to those listed in Fig. 2e, others genes are

also involved in noncanonical signaling (Supplementary Fig. 3A). Notably, we found that $T_{SCM}$ from elderly donors ($n = 4$) exhibited relatively weaker canonical Wnt signatures (Supplementary Fig. 3B), as key molecules such as *CTNNB1, FZD4, NFATC1, RAC1, CDC42,* and *FZD6* were not highly expressed in

**Fig. 2 Heterogeneity of CD4 T$_{SCM}$ cells and Wnt signaling. a** Depletion of T$_{SCM}$ CD4 cells during aging. Freshly isolated PBMCs were collected and stained for flow cytometry. The statistical analysis was performed on unpaired samples (*U* Mann–Whitney test) (**** for *p* < 0.0001). **b** Relationship between naive T-cell subsets during aging. The frequencies of T$_{SCM}$ and naive T cells were compared in young and old individuals (Spearman's rank-order test, *p* < 0.0001, *r* = 0.749). **c** Inflammation and aging. Pro-inflammatory molecules were measured in the plasma of young and older donors (*n* = 99 and *n* = 874, respectively). The statistical analysis was performed on unpaired samples (*U* Mann–Whitney test) (** and **** for *p* < 0.01 and *p* < 0.0001, respectively). **d** Rarefaction of CD31 expressing naive CD4 T cells and T$_{SCM}$ CD4 cells during aging and chronic HIV infection. Staining was performed on freshly collected blood of young (*n* = 28) and elderly donors (*n* = 70). Total CD4 T cells (right *Y*-axis) or CD4 T-cell subsets (left *Y*-axis) were enumerated during aging. Absolute counts were monitored in the peripheral blood of Malaysian cohort of healthy donors (*n* = 10) versus cART-treated HIV-infected patients (*n* = 6). The statistical analysis was performed on unpaired samples (*U* Mann–Whitney test; *, **, ***, and **** for *p* < 0.05, *p* < 0.01, *p* < 0.001, and *p* < 0.0001, respectively). **e** Restoration of CD4 T-cells distribution after successful HIV therapy. Longitudinal follow-up of CD4 T-cells subsets frequency was performed before and 48 weeks after the initiation of cART. The statistical analysis was performed on paired samples (Wilcoxon signed-rank test) (**, ***, and **** for *p* < 0.01, *p* < 0.001, and *p* < 0.0001, respectively). **f** Heterogeneity of CD4 T$_{SCM}$ by high-dimensional single-cell flow cytometry staining. CD4 T$_{SCM}$ cells of 20 donors were concatenated. CD4 T$_{SCM}$ clusters were visualized by phenograph and by a cold to hot heatmap, representing the intensity of each marker. Their distribution during aging was represented by the overlaid populations of CD4 T$_{SCM}$ from young and old patients. **g** Decreased of "RTE-like" CD4 T$_{SCM}$ cluster during aging. The frequency of CD31$^+$PTK7$^+$ T$_{SCM}$ CD4 cells was quantified by flow cytometry. The statistical analysis was performed on unpaired samples (Mann–Whitney, * for *p* < 0.05). Source data are provided as a Source Data file for all figures except (**f**).

T$_{SCM}$ clusters from old donors. The complete analysis of gene signatures within T$_{SCM}$ clusters reveals pathways, which are not only unrelated to Wnt signaling (Supplementary Data File 1) but are also altered during aging.

**Loss of Wnt signaling signature in T$_{SCM}$ CD4 and inflammation.** A comparison of the mRNA libraries of CD4 T$_{RTE}$ and T$_{SCM}$ from young and old donors indicates that these subsets may be inprinted with a pro-inflammatory signature with age (Supplementary Fig. 3C–E; elevated levels of *TNF*, *CD40*, and *NFκB*); this was further validated by nanostring (*p* = 0.0067 and *p* = 0.0165, *n* = 10 for *NFκB2* and *NFκBIZ*, Supplementary Fig. 2A). We further observed that ontology clusters that reflect modulations in activation and functional capacity were differentially enriched in T$_{RTE}$ and T$_{SCM}$ with age (Supplementary Fig. 4A, B). These results led us to postulate that a heightened inflammatory signature, demonstrated by changes in NFκB and TNF signaling for all clusters from older donors (Fig. 3f), could contribute to dysregulated CD4 T$_{SCM}$ homeostasis through altered Wnt signaling.

Together with observations from HIV-infected donors, we found evidence which suggest that heightened levels of inflammation and concomitant changes in the levels of homeostatic cytokines[32–34] could disrupt the genetic signatures of T$_{SCM}$ and their further differentiation. Thus, we examined whether the inflammatory environment—induced by aging or chronic inflammation—interferes with CD4 T-cell development or disrupts T$_{SCM}$ homeostasis. As shown above, CD4 T$_{SCM}$, and T$_{RTE}$ frequencies were positively correlated and could therefore be responsive to overlapping homeostatic mediators. In studying the effect of homeostatic, inflammatory, and effector mediators in healthy and HIV donors (Supplementary Fig. 4C, D), we observed that the expression of inflammatory molecules was negatively associated with the prevalence of CD4 T$_{SCM}$ (*n* = 113, *p* = 0.046, *r* = −0.21 for IL-8; *n* = 23, *p* = 0.0292, *r* = −0.4364 for IL-21; *n* = 99, *p* = 0.0063, *r* = −0.2727 for sCD163; *n* = 113, *p* = 0.0065, *r* = −0.2546 for sCD14, *n* = 98, *p* = 0.0118, *r* = −0.2787 for neopterin). Among the different pro-inflammatory cytokines measured, only IFN-γ was positively associated with CD4 T$_{SCM}$ frequencies (*n* = 76, *p* = 0.0004, *r* = 0.3967). As previously described for IL-7 and naive T cells during HIV infection[32], IL-21 concentration was also negatively associated with percentages of CD4 T$_{RTE}$ (*n* = 23, *r* = −0.5332 and *r* = −0.5953, respectively). This difference in correlation values between T$_{SCM}$ and T$_{RTE}$ suggests a difference in sensitivity to the systemic inflammatory environment, which is also highlighted by the increased expression of activation markers by these subsets (HLA-DR and Ki-67, Supplementary Fig. 4E). We next

investigated how these molecular changes could be dynamically translated into functional or developmental features.

**Functional erosion of T$_{SCM}$ CD4 cells during aging.** In light of our findings, we tested whether T$_{SCM}$-specific functional adaptations occur during aging. While the main transcriptomic signatures of T$_{SCM}$, T$_{CM}$, T$_{EM}$, and T$_{NAIVE}$ cells remain consistent during aging[35], genes encoding transcription factors (*LEF1*, *TCF7*, *TBX21*, *FOXP3*, *IRF4*, *BATF*, *RORC*, and *NOTCH1*), cytokines (*IL-5*, *IL-21*, *IL-17A*, and *TGF-β*), cytokine receptors (*IL7R*, *SOCS1*, *TNF*, and *IL2R*), chemokines (*CCL3*, *CCL4*, *CCL15*, and *CCL20*) and effector molecules (*GNLY*, *GZMB*, and *PRF1*) were modulated (Supplementary Fig. 2A, B) with age. Thus, we measured the ability of young and aged CD4 T-cell subsets to proliferate in response to TCR or homeostatic stimulation. As expected, the highest proliferative potential in healthy young donors was observed in T$_{SCM}$ cells (Fig. 4a), but T$_{SCM}$ cells from older adults showed compromised proliferative capacity (*p* = 0.0205, *n* = 15)—even when compared with T$_{CM}$ cells (Fig. 4b). Moreover, the induction of T$_{SCM}$ proliferation by IL-7, and the secretion of IL-2, IFN-γ, and TNF by proliferating T$_{SCM}$, were impaired in older donors (Supplementary Fig. 5A).

Finally, we studied the expression of three key markers of T-cell differentiation/activation—Ki-67, T-bet, and Eomes (Supplementary Fig. 5B). Here, we found that TCR stimulation of T$_{SCM}$ induced similar T-bet expression (corresponding to their effector functions), but lower Eomes and Ki-67 acquisition than in T$_{CM}$. Moreover, T$_{SCM}$ from FACS-sorted and polyclonally stimulated CD4 subsets from elderly (Fig. 4c) revealed diminished IL-2 and IL-21 secretion (*p* = 0.0101 and *p* = 0.0303, respectively, *n* = 10); the release of effector cytokines (IFN-γ and TNF) (*p* = 0.0415 and *p* = 0.0186, respectively) and pro-inflammatory molecules (IL-17A and CCL20) were also significantly reduced (*p* = 0.0496 and *p* = 0.0062, respectively).

**Preservation of T$_{SCM}$ proliferation is unrelated to clonality.** Since the in vitro functionality of CD4 T$_{SCM}$ was modified with age, we rationalized that the capacity for asymmetric division and proliferation by CD4 T$_{SCM}$ could also be affected in vivo. NOD SCID Gamma (NSG) mice were separately transplanted with human CD4 T$_{SCM}$ that were generated, in vitro, from T$_{NAIVE}$ CD4 precursors from young and old donors (Fig. 4d; Supplementary Fig. 5C). Surprisingly, the engraftment of human CD4 T$_{SCM}$ cells was more efficient for both lymphoid (spleen, *p* < 0.01) and nonlymphoid tissue (lung, *p* < 0.01) when cells were obtained from old rather than young donors (Fig. 4d)—this suggested that intrinsic age-associated factors related to tissue migration or homeostatic function in CD4 T$_{SCM}$ could affect the frequency and

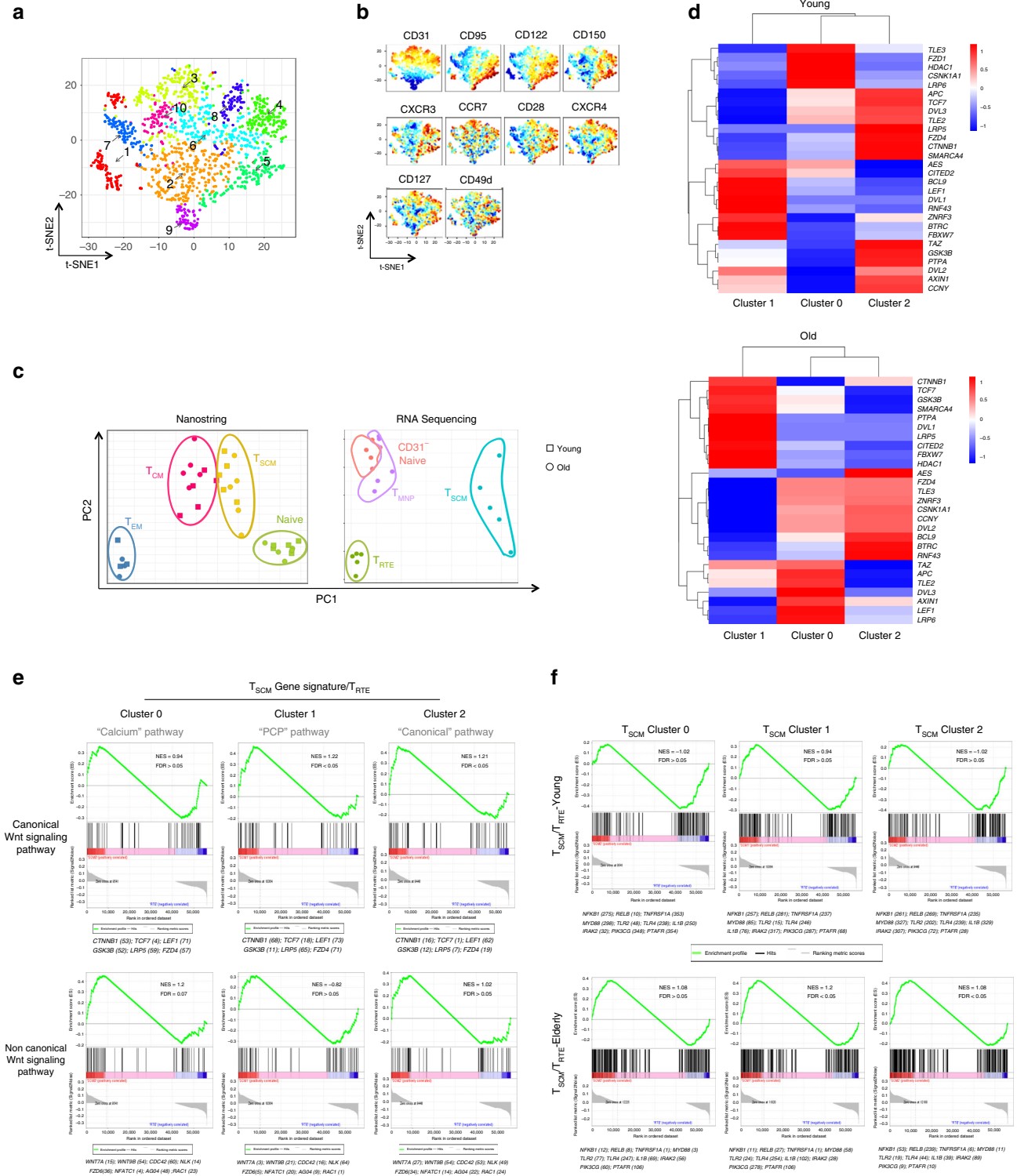

location of peripheral $T_{SCM}$. We tracked the time-sensitive development of transplanted cells (31 days after $T_{SCM}$ transplantation) and observed that the fate of transplanted $T_{SCM}$ cells located within three main cellular clusters, corresponding to $T_{CM}$, $T_{SCM}$, and late transitional memory cells (Supplementary Information). Altogether, our data show that transplanted CD4 $T_{SCM}$ from both young and old donors could persist in their hosts and differentiate into both $T_{TM}$ and $T_{CM}$—albeit with the possible

requirement that they are inducted into a younger noninflammatory environment.

Finally, since TCR diversity is typically lost in T-cell subsets with age[36,37], we wanted to test whether CD4 $T_{SCM}$ are subjected to the the same fate. Here, we fitted RNA-Seq data into a Circos plot to visualize changes in β-chain V–(D)–J rearrangement of $T_{SCM}$ with age (Fig. 4e) through the use of Circos plots. Here, we observed that the greatest TCR diversity was observed in CD4

**Fig. 3 Heterogeneity of the $T_{SCM}$ and differential engagement of Wnt signaling. a** Heterogeneity of naive CD4 T cells revealed by high-dimension flow cytometry. Naive CD4 T cells of eight donors (four young and four old donors), defined as $CCR7^+CD45RO^-CD27^+$, were concatenated and analyzed by t-SNE. Clusters were visualized by phenograph. **b** Phenotype of naive CD4 T-cell clusters. The intensity of fluorescence for each marker was visualized by a normalized heatmap (gradient of increased expression from blue to red). **c** Preservation of specific gene signature of CD4 $T_{SCM}$ cells during aging. Naive, $T_{SCM}$, $T_{CM}$, and $T_{EM}$ CD4 T cells ($n = 5$ for all subsets by age except $T_{EM}$, $n = 3$) were sorted, and analyzed for their gene expression by nanostring. Naive T-cell subsets were sorted as naive, $T_{RTE}$, $T_{MNP}$, and $T_{SCM}$ CD4 cells and analyzed by RNA-seq ($n = 5$). PCA analysis of mRNA expression was performed to evaluate the specificity and preservation of T-cell subsets signature during aging. **d** Heterogeneity of the CD4 $T_{SCM}$ and Wnt/β-catenin signaling pathway. Identification of CD4 $T_{SCM}$ clusters in young and old donors by scRNAseq. Single cells corresponding to CD4 $T_{SCM}$ in all naive CD4 T cells subsets from young ($n = 946$) or old donors ($n = 993$) were analyzed by t-SNE, and clusters were automatically identified. The expression of genes coding for the Wnt/β-catenin pathway was quantified and normalized for each cluster in young and old donors. **e** Canonical and noncanonical Wnt signaling signatures in CD4 $T_{SCM}$ clusters from young donors. The enrichment of gene expression detected in CD4 $T_{SCM}$ clusters for each pathway was calculated in comparison with $T_{RTE}$ signature. Enrichment plot of the gene set reported by GSEA as most enriched among all canonical and noncanonical signaling pathway gene sets (GO:0016055). The profile shows running enrichment score (green curve) and positions of gene set members (black vertical bars) on the rank ordered list of differential gene expression. **f** Inflammatory signature in CD4 $T_{SCM}$ clusters from young and old donors. Enrichment plot of the gene set reported by GSEA as most enriched among all inflammatory gene sets (GO:0006954). The profile was displayed as in (**e**).

$T_{MNP}$—independently of age—and that the loss of heterogeneity in TCR clonotypes was most significant in both CD4 $T_{RTE}$ and $T_{SCM}$ from elderly donors.

**Regulation of $T_{SCM}$ CD4 cells homeostasis: the Wnt/DKK-1 axis.** The dysregulation of the Wnt/β-catenin pathway during aging was further validated at the proteomic level. TCF-1, a main transcription factor within this signaling pathway has been associated with the stemness of CD8 T cells[38–40] and SLAMF-6 (CD352) expression[41,42]. We attempted to validate the relationship between TCF-1 and SLAMF-6 and observed positive correlations between TCF-1 and SLAMF-6 expression only in $T_{RTE}$ and $T_{SCM}$ from young donors ($n = 20$) (Fig. 5a–c); TCF-1 and SLAMF-6 have distinct dynamics during CD4 T-cell differentiation and aging. This observed age-associated decline in TCF-1 expression and decoupling with SLAMF-6 expression could reflect perturbations within the extended Wnt-dependent regulatory network that governs $T_{SCM}$ homeostasis. Such disturbances include an increased frequencies of autoantibodies—indirect predictors of signaling activity and immune response[43,44]—against several protein targets that act in the Wnt/β-catenin pathway with age ($n = 60$ and $n = 93$ for young and older donors, respectively), including CTTNB1 ($p < 0.0001$), GSK3B ($p < 0.01$), TCF-4 ($p < 0.01$), LEF1 ($p < 0.01$), IRF4 ($p = 0.05$), HDAC1 ($p < 0.0001$), and HDAC3 ($p = 0.0064$) (Fig. 5d; Supplementary Fig. 6A). Moreover, plasma concentrations of DKK-1, a natural inhibitor of the Wnt/β-catenin pathway, were increased during aging (Fig. 5e; $p < 0.0001$ and $n = 78$); while levels of SFRP1, a natural activator of Wnt/β-catenin pathway, were reduced ($p < 0.0001$ and $n = 80$). Both circulating levels of SFRP1 and DKK-1 correlated with CD4 $T_{SCM}$ frequencies (Fig. 5f).

Concomitantly, age-associated differences in SFRP1 and DKK-1 activity may interact with the loss of TCF-1 expression to restrict signaling within the canonical Wnt/β-catenin pathway and limit the persistence/expansion of $T_{SCM}$ CD4 cells. We studied this phenomenon in HIV-infected patients, who are often used as a model of inflammatory aging. Higher concentrations of DKK-1 were detected in the plasma of older HIV donors (Supplementary Fig. 6B, $p < 0.0001$ and $n = 97$) and HAART concomitantly reduced inflammation and circulating DKK-1 ($p = 0.0068$ and $n = 12$). These improvements in inflammatory and DKK-1 profiles in treated donors suggested that antiretroviral therapy could promote $T_{SCM}$ differentiation via these mechanisms. We demonstrate further associations between DKK-1 and cellular markers of immune activation (sCD14, $p = 0.0093$, $r = 0.2728$, $n = 90$; IL-26, $p = 0.0245$, $r = 0.2306$, $n = 95$; IDO activity, $p = 0.0008$, $r = 0.3362$, $n = 97$; sCD163, $p = 0.0056$, $r = 0.2795$, $n = 97$) in Fig. 5g. Altogether, the latter strengthens our

hypothesis that the age-related impairments of $T_{SCM}$ homeostasis could be result from meta-inflammation that affects Wnt/β-catenin signaling through DKK-1. Of note, our transcriptomic analysis of the Wnt/β-catenin pathway reveals that *DKK-1* gene expression (as observed in tumors[45]) was only enriched in Cluster 1 $T_{SCM}$ from young donors ("PCP enriched"); *CDH1* (another Wnt/β-catenin inhibitor), which is involved in noncanonical *Wnt* signaling for all donors (Supplementary Fig. 6C), was also dominantly expressed within the $T_{SCM}$ cluster. Among other genes related to Wnt/β-catenin inhibition in older donors, only *SFRP5* and *DACT1* were, respectively, upregulated in Clusters 0 ("PCP like") and 1 ("Wnt/β-catenin like"). The increased DKK-1 activity, is therefore, unlikely to be due to an intrinsic expression of DKK-1 by $T_{SCM}$ of aged donors.

**RTE CD4 T cells are fitter precursors of $T_{SCM}$.** The inhibition of glycogen synthase kinase-3β by TWS119 was shown to promote the in vitro activation of the Wnt/β-catenin pathway in naive T cells, which led to the generation of $T_{SCM}$[9]. Borrowing this approach, we attempted to generate inducible CD4 $T_{SCM}$ ($iT_{SCM}$), and observed that this process was significantly less efficient in older donors ($p = 0.0098$, $n = 20$) (Fig. 6a). Increasing the dosage of TWS119 (10 μM) ameliorated the loss of expansion capacity of CD4 $T_{SCM}$ ($p = 0.007$, $n = 10$)—suggesting that the reported dose of 5 μM may be insufficient to overcome the age/inflammatory-associated inhibition of this pathway. Congruent with the ex vivo phenotype of isolated $T_{SCM}$, $iT_{SCM}$ had a $CD62L^{high}CD45RA^+CD45RO^{-/+}CD95^+$ phenotype. We performed further phenotypic analysis to compare ex vivo and $iT_{SCM}$ in donors from different age groups (Fig. 6b). The increased expansion of $iT_{SCM}$ with higher concentrations of TWS119 was also associated with CD127 recovery ($p = 0.0317$, $n = 5$). Moreover, TWS119-induced CD4 $T_{SCM}$ presented with lower levels of activation markers: CD95 ($p = 0.0013$), CD26 ($p = 0.0079$), CCR5 ($p = 0.0159$), CXCR3 ($p = 0.0391$) and maintained high levels of CCR7 expression compared to control cells ($p = 0.0079$) (Supplementary Fig. 7A); altogether suggestive of a younger phenotype with greater proliferative, functional, and trafficking potential. The phenotypic similarities between $iT_{SCM}$ and ex vivo CD4 $T_{SCM}$, in terms of surface marker expression (Supplementary Fig. 7B), can be corroborated by their compatible genetic signatures[8,15,16,18].

The parallel fates—and ontogenetic proximity—of $T_{NAIVE}$ and $T_{SCM}$ cells suggested that the age-related decline of thymic function is likely to contribute to the loss of peripheral $T_{SCM}$. Furthermore, a decay in the expression of CD31 and PTK7 expression in CD4 T cells from cord blood to young followed by older donors (Fig. 6c), supports a hypothesis that lifelong

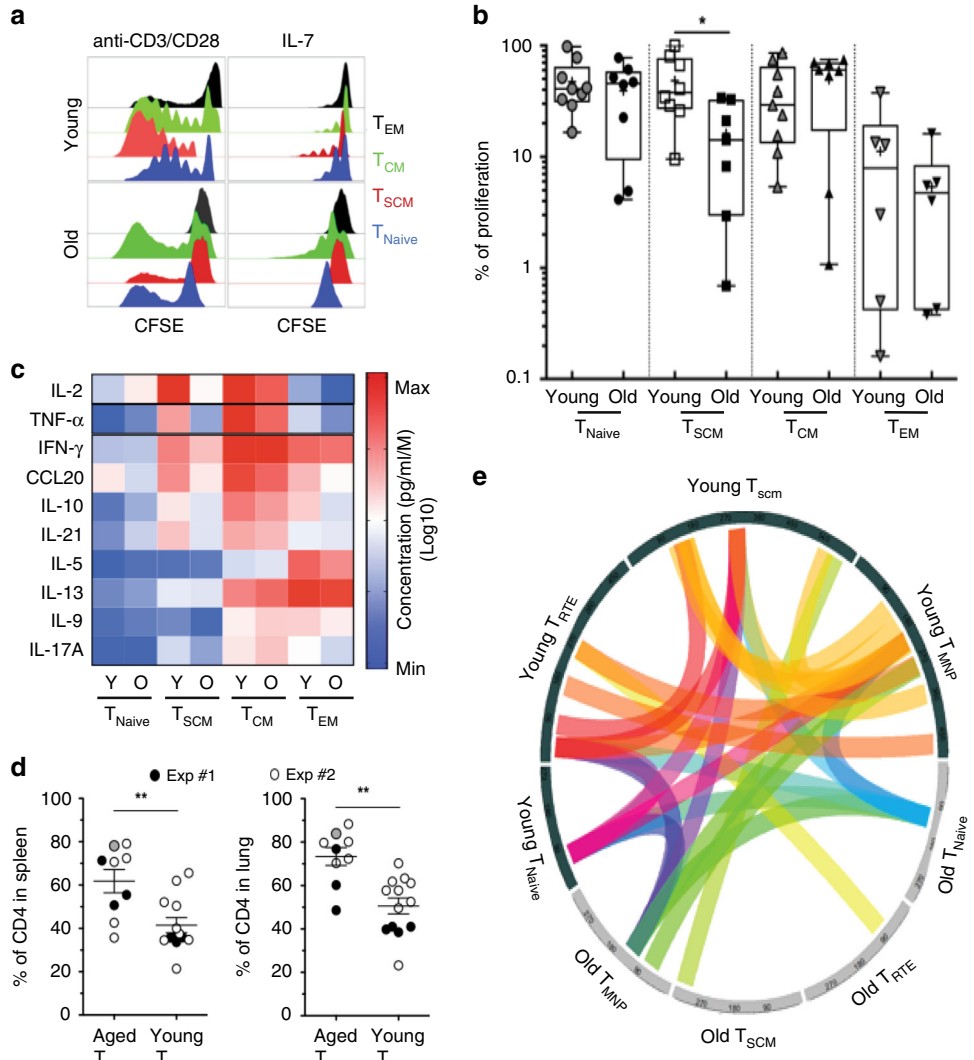

**Fig. 4 Functions of $T_{SCM}$ CD4 cells during aging. a** Proliferation profile of CD4 T-cell subsets during aging. Representative histogram of CFSE dilution from sorted T-cell subsets collected in young or older donors and stimulated with anti-CD3/CD28 microbeads or IL-7 (10 ng/ml) during 5 days. **b** Alteration of proliferative potential of $T_{SCM}$ CD4 cells in response to TCR stimulation as measured in (**a**). T-cells subsets were freshly isolated from blood of young and older donors ($n = 8$ and 9, respectively). The statistical analysis was performed on unpaired samples (*U* Mann–Whitney test) (* for $p < 0.05$). Source data are provided as a Source Data file. **c** Decreased secretion of homeostatic and effector cytokines by $T_{SCM}$ CD4 cells during aging. Sorted CD4 T-cell subsets were polyclonally stimulated with PMA/Ionomycin. The cytokines concentration was represented by an heatmap to visualize the acquisition of effector functions during differentiation and the specific signature associated with aging ($n = 6$ for young and old donors). Source data are provided as a Source Data file. **d** Increased engraftment of human $T_{SCM}$ CD4 cells from aged donors in humanized NOD SCID gamma chain (NSG) mice. $T_{SCM}$ CD4 cells from young ($n = 2$) or old donors ($n = 2$) were differentiated from T-naive precursors, expanded in vitro before their xenotransplantation into NSG mice ($n = 13$ and $n = 9$, respectively). At 21 (Exp#1) or 28 (Exp#2) days after the human $T_{SCM}$ CD4 cells transfer, animals were killed and euthanized by $CO_2$, and tissues were collected (spleen, lungs). $N = 2$ independent experiments were performed and labeled by the color code on the graph (filled circles: Exp#1; open circles: Exp#2; filled gray circle represents the mouse with GVHD signs and killed at day 16). The statistical analysis was performed on unpaired samples (*U* Mann–Whitney test) (** for $p < 0.01$). **e** Reduced CDR3 diversity in naive CD4 T cells. Naive T-cell subsets were sorted as naive, $T_{RTE}$, $T_{MNP}$, and $T_{SCM}$ CD4 cells. The extraction of mRNA was performed just after T-cell sorting, and analyzed by RNA-seq. CDR3 composition was compared between cell subsets and during aging. A connective arc represented high degree of homology (80%) between CDR3 sequences during differentiation and aging.

immune stimulation contributes to CD31 shedding[46]. We also observed a positive correlation between the frequencies of $T_{RTE}$ cells (defined as PTK7+CD31+) at day 0 and the frequencies of induced $T_{SCM}$ at day 7 (Fig. 6d), supporting the idea that $T_{SCM}$ could be derived from $T_{RTE}$ cells. This hypothesis was evaluated in silico by STEM analysis of naive CD4 T-cell subsets (Supplementary Fig. 7C, D). The dynamics of gene expression between $T_{RTE}$ and $T_{SCM}$ CD4 cells was a more plausible outcome than comparisons of other putative ontogenic pathways. Next, Monocle was used to generate pseudotime from single-cell RNA-

seq data and determine CD4 T-cell differentiation trajectory- our transcriptome supports a model where $T_{RTE}$ could differentiate into $T_{SCM}$ and CD31neg naive CD4 T cells (Supplementary Fig. 7E).

Due to the overlapping features of $T_{RTE}$ and $T_{SCM}$ with age, particularly the loss of Wnt/β-catenin pathway activity with age— we speculated that high CD31 expression could reduce the threshold of naive CD4 T cells to differentiate into $T_{SCM}$ via the Wnt/β-catenin activation (Fig. 6e). We observed that, independent of donor age, 5 μM of TWS119 concentration could

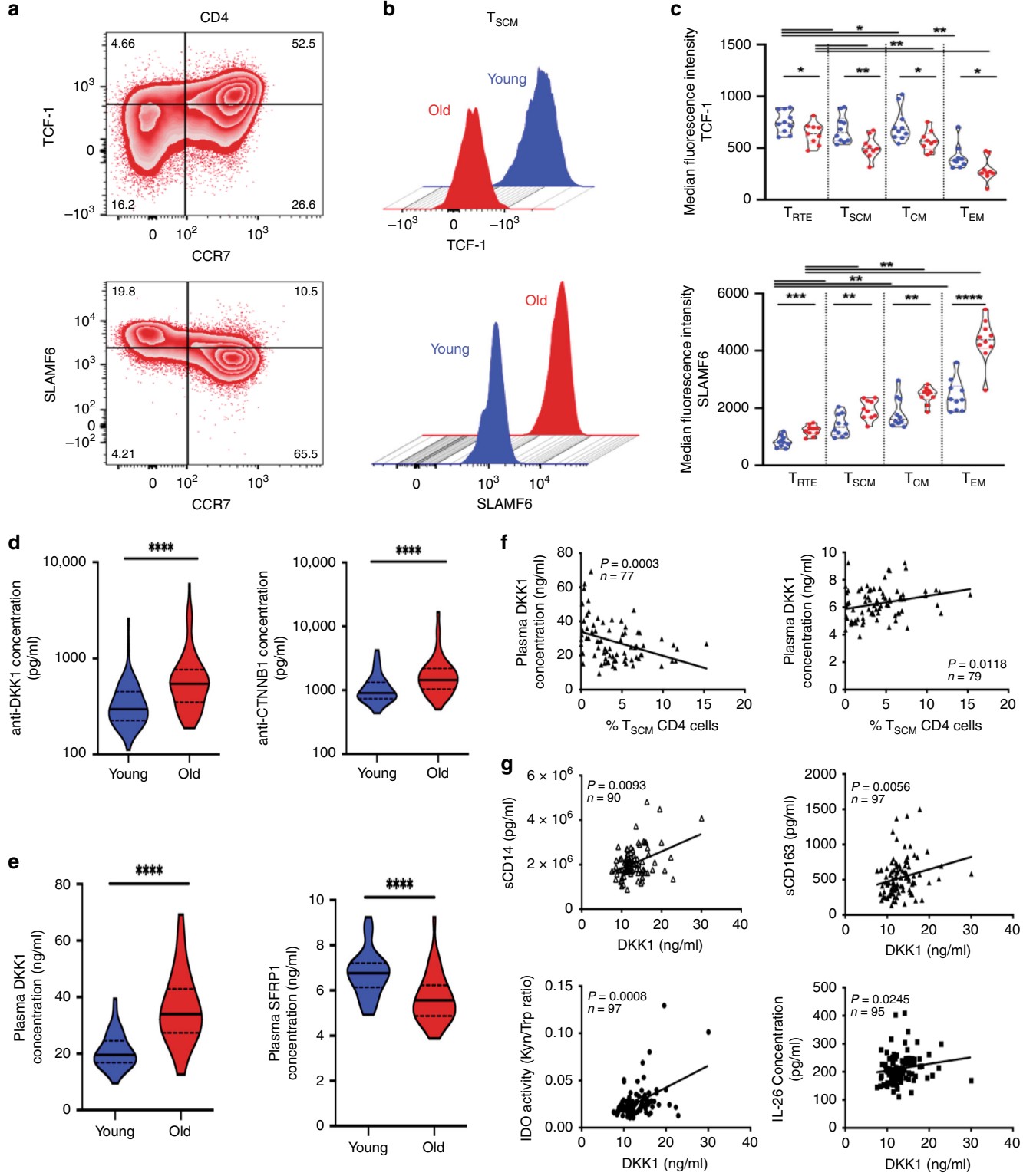

efficiently mediate the induction of FACS-sorted CD31$^{high}$ naive CD4 T cells into T$_{SCM}$ CD4 cells (iT$_{SCM}$) (Fig. 6f, $p = 0.0008$ and $n = 9$). Conversely, CD31$^{neg}$ naive CD4 T cells were resilient to conversion at 5 μM concentration of TWS119, but responded when 10 μM was used ($p = 0.0007$ and $n = 9$). In monitoring the expression of CD45RA, CCR7, CD127, and CD27 on CD31$^{neg}$ naive CD4 T cells, we also obtained evidence that TWS119 could promote a more "naive" phenotype in CD31$^{neg}$ naive CD4 T cells at higher dosage (Fig. 6g).

**Metabolism of T$_{SCM}$ CD4 cells during aging**. Several groups have described the age-related hypermethylation of genes (*IFNG, CCR7, CD27*, etc.) that result in functional changes in naive T-cell behavior[22,47,48]. Guided by these principles, we studied whether groups of genes were simultaneously modulated (grouped by behavioral profile) with progressive T-cell differentiation (i.e., from T$_{RTE}$ to T$_{TMNP}$) using STEM analysis (Supplementary Fig. 7C). We identified the top canonical pathways that were associated with T-cell differentiation in different age groups, as

**Fig. 5 Regulation of T$_{SCM}$ CD4 cells homeostasis during aging. a** TCF-1 and SLAMF-6 expression in CD4 T cells. Representative zebra plots of SLAMF-6 and TCF-1 staining in CD4 T cells from a representative old individual. **b** TCF-1 and SLAMF-6 expression in T$_{SCM}$ CD4 cells during aging. Representative overlaid histograms plots of SLAMF-6 and TCF-1 expression in gated T$_{SCM}$ CD4 cells from young ($n = 10$) and older ($n = 10$) individuals. **c** Decreased expression of TCF-1 during CD4 T-cell differentiation and aging. The median fluorescence intensity of TCF-1 was measured in T-cell subsets. The statistical analysis was performed on paired ($n = 20$, Wilcoxon signed-rank test) or unpaired samples ($n = 20$, Mann–Whitney; *, **, ***, and **** for $p < 0.05$, $p < 0.01$, $p < 0.001$, and $p < 0.0001$, respectively). Source data are provided as a Source Data file. **d** Alternative activation of Wnt/β-catenin pathway by DKK-1 during aging. Cryopreserved plasma was used to measure autoantibodies directed against molecules involved in the Wnt/β-catenin pathway ($n = 93$ and $n = 60$ in young and old, respectively). The statistical analysis of immunone protein array data was performed on unpaired samples ($U$ Mann–Whitney test, **** for $p < 0.0001$). Source data are provided as a Source Data file. **e** Modulation of the natural inhibitor and agonist of the Wnt/β-catenin pathway during aging. The plasmatic concentration of DKK-1 and SFRP1 was measured directly by ELISA ($n = 43$ and $n = 37$ in young and old donors, respectively). The statistical analysis was performed on unpaired samples ($U$ Mann–Whitney test, **** for $p < 0.0001$). Source data are provided as a Source Data file. **f** Regulation of T$_{SCM}$ CD4 cells by DKK-1 and SFRP1. The frequency of T$_{SCM}$ CD4 cells correlated negatively or positively with the systemic concentration of DKK-1 and SFRP1, respectively ($p = 0.0003$ and $p = 0.0118$) ($n = 77$). The correlations were calculated with the Spearman's rank-order test. Source data are provided as a Source Data file. **g** Inflammation and DKK-1 plasma levels. The concentration of DKK-1, sCD14, sCD163, and IL-26 was measured directly by ELISA. Plasma levels of tryptophan and L-kynurenine were measured by LC-MS/MS. The correlations were calculated with the Spearman's rank-order test. Source data are provided as a Source Data file.

well as the respective upstream regulators that were responsible for the observed phenotype. T-cell differentiation was associated with different metabolic profiles between the young and old, suggesting differences in energy management with age. For example, in young donors, anabolic pathways such as diacylglycerol and phosphatidylglycerol biosynthesis were modulated with T-cell differentiation (Cluster 11), while genes involved in catabolic processes such as oxidative phosphorylation were highlighted in older donors. We also observed differences in Cluster 5, where major shifts in the regulation of acid biosynthesis (glutamine, serine, and glycine) and glycogen biosynthesis were observed in young and elderly donors, respectively (Cluster 5; Supplementary Fig. 7D). In examining the signaling targets that are altered with progressive naive CD4 T-cell differentiation, we observed possible alterations in the activation of specific signaling and metabolic pathways (*RhoA, Sirtuin, mTOR,* and *MYC*). These canonical pathways are regulated by upstream regulators, which were distinct for each age group within the same clusters of concordantly regulated genes. We detected the naive T-cell differentiation could be differentially guided by the influence of homeostatic cytokines (*STAT5A*) as well as by the environment through the alternate engagement of viral sensors (*IRF3, IFNB1,* and *IL12B*) within the two age groups. For example, the energetic requirements for the development (*TSC22D3, POU2F2*), differentiation, or acquisition of effector functions (*TSC22D3, IRF3,* and *LEPR* for Th17 cells) are specific to each CD4 T-cell subset. The priming and differentiation of naive CD4 T cells are thus coupled with specific changes in gene expression and metabolic gene signature during aging.

**Polarization of T$_{SCM}$ CD4 cells during aging**. In addition to phenotypic and molecular dissimilarities, we endeavored to identify morphological and structural changes that may develop in T$_{SCM}$ with age as a possible response to the differential engagement of Wnt signaling pathways (PCP in particular and possibly due to DKK-1) with age—as any visible differences in their surface architecture could also help to explain differences in T$_{SCM}$ behavior. We investigated on the potential implication of the Wnt pathway in the CD4 T$_{SCM}$ polarization. The atypical expression of *CDC42* in Wnt/β-catenin cluster in T$_{SCM}$ from old donors (Supplementary Fig. 3B) led us to propose that the orchestration of cytoskeletal events, including the distribution of proteins associated with polarity, might be impaired in the elderly. However, TCR-mediated stimulation led to the expected unipolar recruitment of Cdc42 in CD4 T cells from young donors, but such polarization was infrequent in aged donors (Supplementary Fig. 8A, B). The latter was particularly the case for CD31$^-$ naive

CD4 T cells, but this trend was also observed for T$_{CM}$ and T$_{SCM}$ cells, albeit absent in CD31$^{high}$ naive CD4 T cells (T$_{RTE}$).

Due to the distinct polarization profiles of naive CD4 T-cell subsets, we sought to determine whether the main regulator and source of chemical energy, i.e., the mitochondria, behaved differently in CD4 T$_{SCM}$ cells during aging[49,50] (Supplementary Fig. 8C). We observed a reduction in the average mitochondrial volume (but not of mitochondria numbers, Supplementary Fig. 8D) in T$_{SCM}$ CD4 cells in the elderly as compared with young donors ($p < 0.05$) (Supplementary Fig. 6D).

Overall, these multidimensional changes in the patterns of T$_{SCM}$ gene and protein expression advocate strongly for the argument that systemic changes in the frequency and function of T$_{SCM}$ cells in the elderly could to a large extent, be explained by disturbances to the cellular environment (summarized in Fig. 7).

## Discussion

Naive CD4 T cells are a heterogeneous population in terms of gene expression, phenotype, and function, and are divided into subclasses that respond differently to external signals—such as chronic infection, vaccination, and inflammaging. The various inflammatory contexts examined in this study demonstrate that CD4 T$_{SCM}$ and their progenitors are sensitive to the external environment. Immune activation induced by persistent infections such as HIV and CMV may imprint specific behavior to CD4 T$_{SCM}$ cells. The clonal expansion of differentiated virus-specific T cells may also indirectly shape T-cell repertoire and therefore limit the responsiveness to future challenges. In this study, we demonstrate a quantitative and qualitative (proliferation, effector function) defect in CD4 T$_{SCM}$ cells during aging and chronic infections. We also provide multiple evidence to show that persistent inflammation could indeed interfere with the functioning of these subsets at the single-cell level—these changes were accompanied by changes to Wnt/β-catenin gene expression, and associated with specific proteomic and metabolic signatures. Essentially, while all naive T cell can differentiate, the most likely precursors of CD4 T$_{SCM}$ cells appear to reside in the T$_{RTE}$ compartment, which is itself severely compromised in the contexts of aging (reduced thymopoeisis, inflammation) and chronic infections (clonal expansion of memory T cells, which may compete for space and resources). Immune activation, TLR stimulation, and the binding of innate viral sensors may also activate putative upstream TFs that act to orchestrate biased T-cell differentiation in the elderly, possibly via DKK-1 modulation[51]. Inflammation could thus affect CD4 T$_{SCM}$ cells directly and indirectly even at the RTE precursor stage.

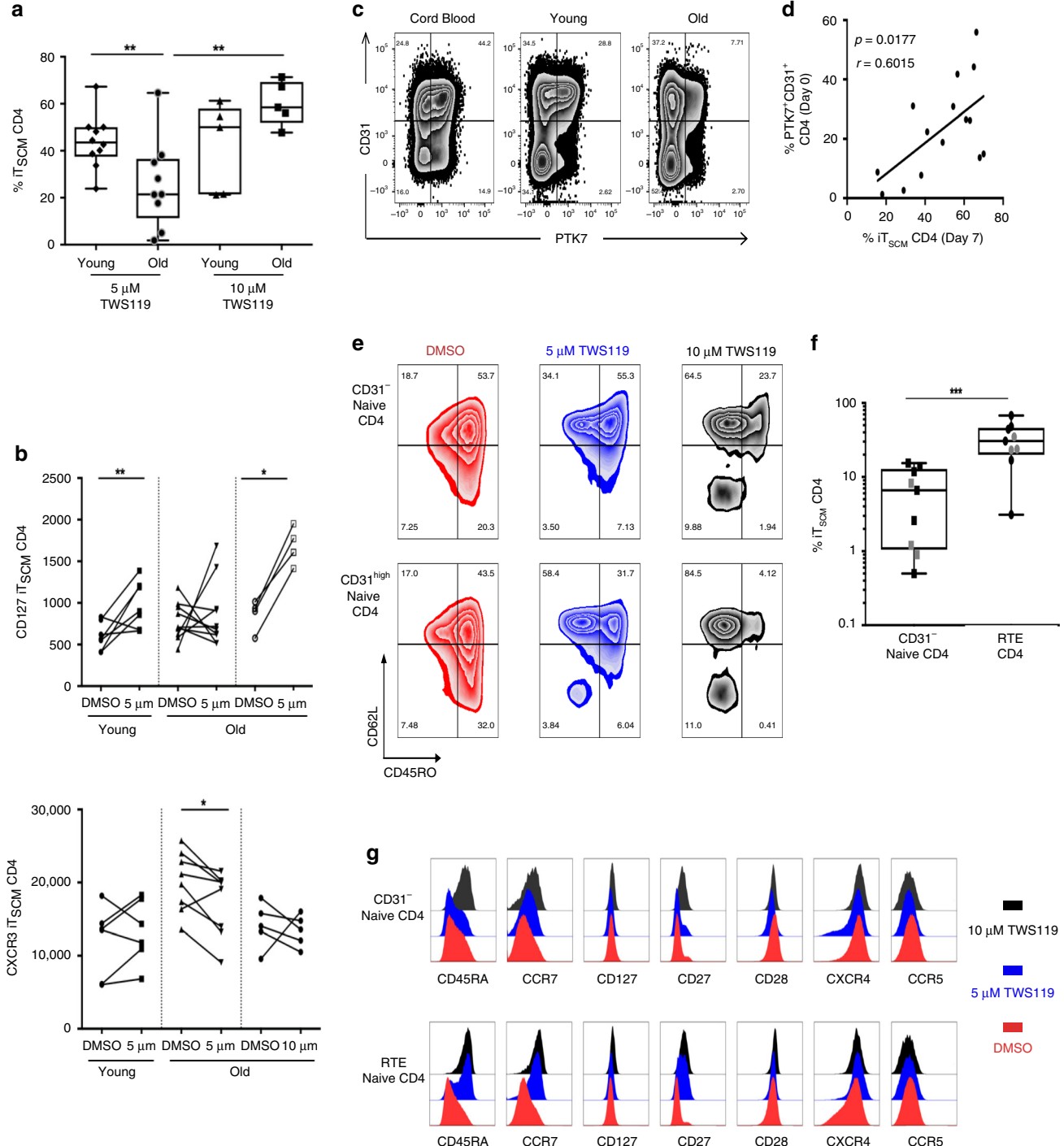

In describing the extent of CD4 $T_{SCM}$ depletion that accompanies aging and chronic inflammation induced by HIV infection, and linking these phenomena to immune activation and the Wnt/β-catenin pathway in this phenomenon—we propose that modulation in the gene expression of $T_{SCM}$ cells, which manifest most strikingly in their impact on metabolic and signaling pathways—could be significantly explained by alterations in the inflammatory environment (Fig. 7). This age-dependent signature of $T_{SCM}$ could contribute to sub-optimal $T_{SCM}$ differentiation and increased susceptibility to cellular senescence via a mechanism that is independent of antigenic source and linked to the nature of the inflammatory environment. Thus, we demonstrate that the sub-optimal immune response that is observed during aging and

HIV infection may evolve partly from the loss of CD4 $T_{SCM}$ heterogeneity through altered Wnt signaling engagement.

Our conclusions are further substantiated by observations that CD8 $T_{SCM}$ depletion is been associated with disease progression, in the contexts of HIV[52,53] or symptomatic *Trypanosoma cruzi* infections[54]; the generation of antigen-specific CD4 $T_{SCM}$ cell with more diverse effector molecules (Granzyme A, K) and chemokines (CCR5, CCR6) profiles during acute *Mycobacterium tuberculosis* infection also contributes to a better prognosis[55]. The importance of having adequate CD4 $T_{SCM}$ heterogeneity is also supported by the results from the RV144 prime-boost HIV-vaccine trial[56]. Similarly, a selective depletion of CCR5+ $T_{SCM}$ CD4 cells was observed in pathogenic but not in non-pathogenic

**Fig. 6 Regulation of T$_{SCM}$ CD4 cells homeostasis during aging. a** Hyporesponsiveness of the Wnt/β-catenin pathway during aging. The frequency of specifically TWS119-induced T$_{SCM}$ CD4 cells was evaluated by flow cytometry, and calculated after the subtraction of nonspecific DMSO-induced T$_{SCM}$ CD4 cells frequencies. The statistical analysis was performed on unpaired samples (U Mann–Whitney test, ** for $p < 0.01$). Source data are provided as a Source Data file. **b** CD127 expression characterizes induced-T$_{SCM}$ CD4 cells. The phenotype of induced-T$_{SCM}$ CD4 T cells was performed at day 7. The statistical analysis was performed on paired samples (Wilcoxon signed-rank test) (* and ** for $p < 0.05$ and $p < 0.01$, respectively). Source data are provided as a Source Data file. **c** Identification of recent thymic emigrants in cord blood or in peripheral blood of young and old donors by flow cytometry. T$_{RTE}$ were defined as PTK7$^+$CD31$^+$CD4 T cells. **d** Induction of T$_{SCM}$ CD4 cells from T$_{RTE}$ cells. The correlations between ex vivo T$_{RTE}$ and in vitro TWS119-induced T$_{SCM}$ CD4 cells at day 7 were calculated with the Spearman's rank-order test ($n = 15$). Source data are provided as a Source Data file. **e** T$_{SCM}$ CD4 cells induction depends on CD31 expression in naive CD4 T cells. Flow cytometry staining of TWS119 dose-dependent induced-T$_{SCM}$ CD4 T cells derived from CD31$^-$ or CD31$^{high}$ naive CD4 T cells. **f** Increased potential of CD31$^{high}$ naive to T$_{SCM}$ CD4 cells differentiation. Frequencies of T$_{SCM}$ CD4 cells derived from CD31$^-$ or CD31$^{high}$ naive CD4 T cells in response to the stimulation of the Wnt/β-catenin pathway with low dose of TWS119 (5 μM). Young and old donors are represented with black and gray symbols, respectively. The statistical analysis was performed on paired samples (Wilcoxon signed-rank test) (***for $p < 0.001$). Source data are provided as a Source Data file. **g** Phenotype of induced T$_{SCM}$ from T$_{RTE}$ and CD31$^{low}$ naive CD4 T cells. Flow cytometry staining of induced-T$_{SCM}$ CD4 cells derived from CD31$^-$ or CD31$^{high}$ (T$_{RTE}$) naive CD4 T cells. Histograms represented overlaid expression of individual markers after 7 days of culture in presence of vehicle alone (DMSO) or Wnt/β-catenin stimulating drug (TWS119) at 5 or 10 μM.

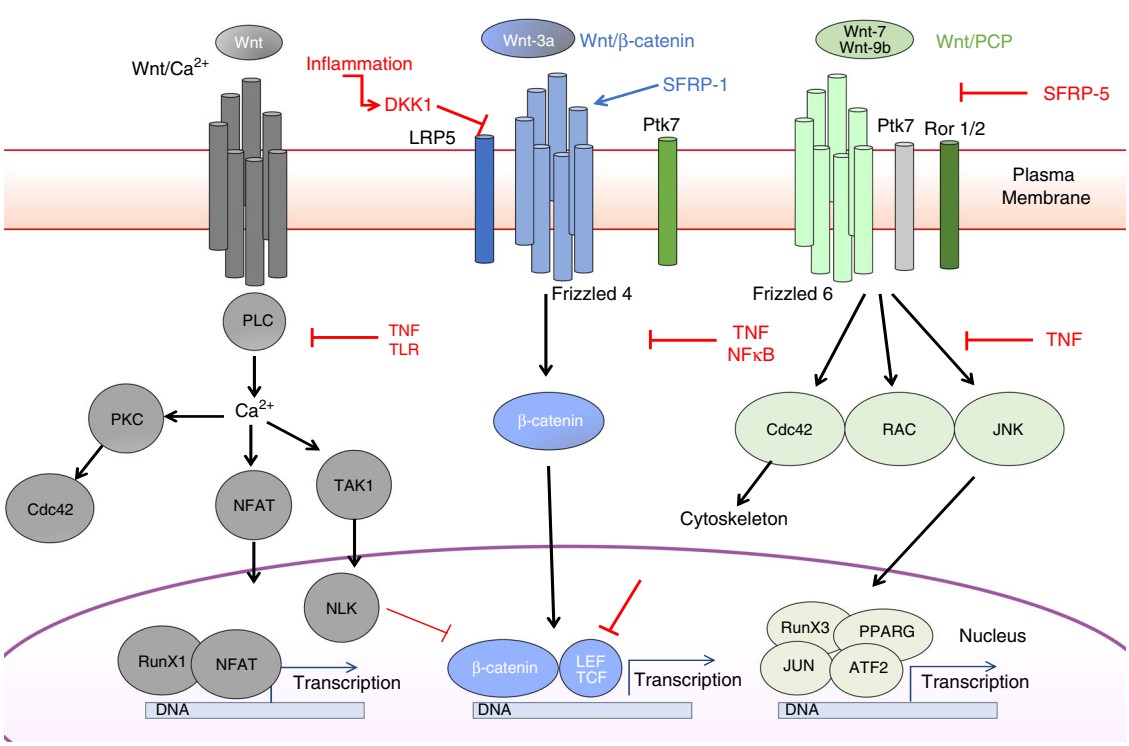

**Fig. 7 Wnt Signaling pathway in CD4 T$_{SCM}$ during aging and inflammation.** Graphical summary of key molecules involved in the specific signature of individual T$_{SCM}$ cluster identified by scRNAseq gene expression analysis. Gray, blue, and green boxes represented highlighted molecules in Wnt/calcium, Wnt/β-catenin, and Wnt/PCP, respectively. Red arrows and characters summarized the alterations of Wnt signaling observed in aged donors and/or chronic inflammation described in this study.

simian immunodeficiency virus infections[57]. The loss of these memory precursors could be due to HIV replication in CD4 T$_{SCM}$, which in turn affects the ability of the host to repopulate the entire library of T cells and to control current and further infections[58].

Whether the alternate engagement of canonical or non-canonical Wnt/β-catenin signaling contributes to heterogeneous clusters of CD4 T$_{SCM}$ warrants further investigation. The ambiguous role of the Wnt pathway activation in HIV replication[59] or in the establishment of HIV latency could result from differences in how T$_{SCM}$ are distributed within the clusters described in this study. Alternatively, HIV prognosis may be affected by the availability of DKK-1 or virus-derived molecules like HIV TAT and YAP that could interact with Wnt signaling pathways. The dynamic flux of T$_{SCM}$ has been recently demonstrated by tracking genetically modified T$_{SCM}$[60], and using stable isotopes and heavy water[61], which supports a model where the T$_{SCM}$ reservoir is continuously replenished and maintained by continuous T$_{SCM}$ turnover in the absence of any telomeric erosion due to the active expression of telomerase. However, the kinetics of human T$_{SCM}$ also follows a dichotomous distribution, with two sub-populations described as either short- or long-lived (5 months versus 9 years, respectively)[11]. These segmentation contributes to three speculations: (i) the alternate Wnt pathway engagement may affect the longevity of these different sub-populations, (ii) these short- and long-lived-T$_{SCM}$ may arise from T$_{NAIVE}$ and CD31$^{high}$ naive T cells, respectively, and (iii) T$_{SCM}$ heterogeneity may be driven by the type of antigen. The integration of environmental signals by T$_{SCM}$ and their proliferative capacity could trigger homeostatic adjustments that preserve

T-cell compartmentalization[57]. The age-associated decline of $T_{SCM}$ cells may therefore compromise this plasticity and the entire memory immune capacity.

Next, the diversity and clonal availability of $T_{SCM}$ were also found to be crucial for generating recall responses[62] and for the long-term maintenance of adult T-cell leukemia[63]. However, CDR3 diversity in $T_{SCM}$ is reduced during aging and, in the absence of cognate epitopes, $T_{SCM}$ persistence depends heavily on homeostatic proliferation (mediated through IL-7, IL-15, or IL-21) and the differentiation of scarce $T_{RTE}$[11]. An increase in IL-7 levels was further correlated to the loss of naive[64] and CD4 $T_{RTE}$ cells[65], which are both predictive of poor HIV prognosis[64–66]. Furthermore, co-infection with HCV presents an additional burden that contributes to the loss of $T_{RTE}$[67]. The reduced survivability and persistence of $T_{RTE}$ and $T_{SCM}$ CD4 cells in the elderly and during HIV infection are likely to be related to both the increased presence of inflammatory cytokines and homeostatic IL-7 levels. In support of this hypothesis, we observed differences in CD127 expression within $T_{SCM}$ clusters that strongly express molecules associated with canonical Wnt/β-catenin signaling. CD127 is considered as a marker of long-term memory[68]. Here, we show this role could evolve through the promotion of Wnt/β-catenin signaling, which leads to enhanced CD4 $T_{SCM}$ proliferative capacity. The loss of CD127 has been further described as a hallmark effect of inflammation, and we confirm this through the context of HIV-associated immune aging. Moreover, some studies have described an increased infectivity of CD4 $T_{SCM}$[58,69] and $T_{RTE}$ by HIV (or SIV in the case of the Rhesus Macaque) in progressors[57]. A preserved $T_{RTE}$ compartment is also associated with higher CD4 nadir[66]. That HIV could gain an evolutionary advantage by undermining CD4 $T_{SCM}$ and $T_{RTE}$ function suggests their importance in the control of viral replication. Moreover, the reconstitution of CD4 $T_{SCM}$ accompanies successful HAART administration—whether this is a cause or effect of successful HIV control warrants further investigation.

The other striking observation in this study was the increased hyporesponsiveness of the Wnt/β-catenin pathway and the concomitant loss of active Wnt/β-catenin genetic signature at the single-cell level during aging and HIV infection. In addition to driving the $T_{SCM}$ differentiation via the route of CD31$^{high}$ CD4 T cells, stimulation of the Wnt/β-catenin pathway with high dosage agonist promoted the acquisition of a CD4 $T_{SCM}$ phenotype even in CD31$^{-}$ naive CD4 T cells, which often possess a homeostatic proliferation history. While the resistance of total naive T cells to i$T_{SCM}$ differentiation in aged donors was linked to lower $T_{RTE}$ frequencies, we observed that $T_{RTE}$ were most pliant to the Wnt/β-catenin pathway stimulation, since they responded regardless of the donor's age, and with minimum agonist dosage. The preservation (or acquisition) of CD127 on naive CD4 T cells was also found to be a reliable indicator of the ease of i$T_{SCM}$ induction. Thus, data from these experiments suggest that the pliability of CD4 $T_{RTE}$ to $T_{SCM}$ differentiation erodes progressively from the time when CD4 $T_{RTE}$ egress from the thymus and that such a phenomenon may be due to alterations in Wnt/β-catenin signaling. Importantly, our results show that there is clinical potential in targeting Wnt/β-catenin signaling to promote the in vivo genesis of CD4 $T_{SCM}$.

Our data consistently reveal an age- or inflammation-dependent dysregulation in the balance of natural agonists and antagonists (DKK-1/SFRP1) of the Wnt/β-catenin pathway and an increased prevalence of autoantibodies against members of canonical Wnt/β-catenin pathway signaling (CTTNB1, GSK3B, IRF4, and HDAC1). All these factors could further contribute to the hyporesponsiveness of this pathway in CD4 $T_{SCM}$ from elderly donors, which dampens downstream T-cell functions.

That loss of CD4 $T_{SCM}$ integrity can be mediated through compromised Wnt/β-catenin signaling can be supported by the observed overexpression of DKK-1 in various cancers[45,70] and by Tregs in the contexts of autoimmune disease and colitis[71]. It was suggested that the inhibitory nature of DKK-1 can be directly influenced by a tumor suppressor gene or indirectly via the induction of myeloid-derived suppressor cells. Accordingly, DKK-1 has been proposed as a target for immune therapy and anti-DKK-1 vaccination was shown to strengthen antitumor immunity[72].

Finally, the aberrant morphological changes in CD4 naive, $T_{SCM}$, and $T_{CM}$ subsets during TCR engagement are indicative that elderly T cells are unable to orchestrate appropriate physiological responses to critical signaling events. Among immature CD4 T-cell subsets from the elderly, spatial organization following TCR engagement was only preserved in $T_{RTE}$. A similar characteristic was observed in HSCs, and this phenomenon was closely related to the noncanonical engagement of the Wnt/β-catenin pathway[73]. Cdc42, a Rho GTPase, is involved in the immunological synapse formation and involved in signal transduction, acto-myosin organization, cell proliferation, and Wnt/PCP and Wnt/Ca$^{2+}$ noncanonical pathways[74,75]. Defective Cdc42 expression, the observed loss of polarity and mitochondrial regulation may therefore contribute to the reduced fitness and metabolic activity of CD4 $T_{SCM}$ cells during aging.

The paucity of $T_{RTE}$ in elderly—driven both by thymic involution and immune activation—results in a ticking time-bomb scenario where responses to new and repeat infections are increasingly crippled by the loss of memory T-cell precursors and naive T-cell heterogeneity. Nevertheless, our data suggest that modulation of the Wnt/β-catenin pathway by introducing high levels of agonist (or better agonists) may be helpful in delaying or preventing the latter scenario; this is an attractive proposal as alternative methods (such as thymic rejuvenation) have been shown to result in adverse effects. The importance of CD4 $T_{SCM}$ in promoting survival and immune competency remains largely unexplored, since most studies have focused on CD8 $T_{SCM}$. In the CD8 context, $T_{SCM}$ have been shown to be important for preventing GVHD, promoting antitumor immunity[8,13] and is required for mounting adequate responses to a plethora of pathogens. The stemness of $T_{SCM}$ CD8 T cells also correlates with the long-term persistence of yellow fever -specific CD8 $T_{SCM}$ during successful vaccination[76]. With regard to CD4 function, our data support the idea that the preservation of CD4 $T_{SCM}$ frequencies is also important for immune competency—particularly in the context of viral control. Thus, promoting CD4 $T_{SCM}$ differentiation could, likewise, be relevant to enhancing vaccine efficacy; this is logical as $T_{SCM}$ constitute an important reservoir for the development of memory T-cell responses. Altogether, our data suggest that strategies that reduce the exposure of CD4 T cells to systemic inflammation may be instrumental to preserving $T_{SCM}$ immunocompetency.

## Methods

**Donors and sample preparation**. Blood was collected from participants of the Singapore Longitudinal Aging Study (SLAS) cohort. Characteristics of the SLAS cohort are detailed in the Supplementary Table 2 and in our previous publication, related to physical frailty[77]. Blood was collected into BD Vacutainer CPT Cell Preparation tubes with Sodium Citrate (BD Biosciences, San Jose, CA, USA). After centrifugation at 300 rcf for 20 min at room temperature, plasma and peripheral blood mononuclear cells (PBMCs) were isolated. Plasma was stored at −80 °C before use. PBMCs were frozen in 90% fetal bovine serum (FBS) containing 10% DMSO and stored in liquid nitrogen.

The study has been approved by the National University of Singapore-Institutional Review Board 04–140 and all participants gave informed consent. Young donors were recruited at the National University of Singapore. The study has been approved by the Ethics Committee of the NUS-IRB 09-256. All study participants provided informed written consent.

Blood was collected from HIV-infected individuals attending the University Malaya Medical Centre (UMMC), Malaysia. Data on HIV-specific characteristics including HIV RNA, CD4 T-cell counts, antiretroviral drug history, and history of co-infections were obtained from patient medical records. The study was approved by the hospital institutional review board for Malaysian HIV-infected patients (MEC 975.6).

All experiments using human buffy coats were approved by the Humanitas Clinical and Research Institute IRB (approval 28/01/2016).

**Animal studies**. All experiments using mice were conducted upon the approval of the Italian Ministry of Health (protocols 256/2015-PR). The permission to perform animal experiments was granted by the Italian Ministry of Health. NOD.Cg-$Prkdc^{scid}$ $IL2rg^{tm1Wjl}$/SzJ (NSG) mice (Jackson Laboratories) were bred in specific-pathogens-free (SPF) conditions.

**Screening of serum**. All serum samples of the entire SLAS cohort were first tested for IgG antibodies against *H. pylori*, EBV, VZV, HSV1, HSV2, CMV, by anti-enzyme-linked immunosorbent assay (ELISA) using commercial test kits (Virion \Serion, Germany) according to the manufacturer's recommended procedure. IgG antibodies against CHIKV and DENV were detected by ELISA from Abcam (USA) and Panbio (USA), respectively.

**Phenotyping of PBMCs**. Cell phenotyping was performed by flow cytometry on 23 fresh PBMC samples from young donors and 78 fresh PBMC samples from older donors. For each staining, $1 \times 10^6$ PBMCs were used. Lymphocytes were gated based on FSC/SSC profile and doublets/dead cell exclusion.

Absolute cell count was performed by flow cytometry on freshly collected blood of ten healthy donors and six HAART HIV-infected patients. For each staining, 100 μL of blood was used in Trucount tubes. After doublets/dead cell exclusion, lymphocytes were gated based on FSC/SSC profile and CD45 expression.

The antibodies are listed in Supplementary Table 3. Flow cytometry was performed on an LSR Fortessa Cell Analyzer (BD Biosciences), and automatic compensation was applied.

We used BDSymphony A5 (BD Biosciences) to perform high-dimensional single cells immunophenotyping and characterize the heterogeneity of naive and $T_{SCM}$ CD4 cells from 2 million frozen PBMCs.

Flow cytometry of HIV participants was performed on a BD FACS Celesta (BD Biosciences) at University of Malaya and automatic compensation was applied.

**Flow cytometry functional assay**. Proliferation: CFSE-stained sorted CD4 T-cell subsets were stimulated with anti-CD3/CD28 microbeads or IL-7 during 5 and 7 days, respectively. Proliferation was measured through the dilution of CFSE. Proliferation/ICS: after 5 days of TCR stimulation, CFSE-stained sorted CD4 T-cell subsets were re-stimulated with PMA/Ionomycin (1 μg/ml and 100 ng/ml, respectively) during 4 h to detect the secretion of cytokines by proliferating CD4 T cells. Brefeldin A and Monensin (eBioscience) were added during the final 2 h of incubation. For the list of antibodies used, refer to Supplementary Table 3. Flow cytometry was performed on an LSR Fortessa Cell Analyzer (BD Biosciences).

**Multiplex analytes screening**. Cell sorting was performed with a FACS Aria III (BD Biosciences) on 12 samples according to CCR7, CD27, CD31, CD45RO, CD49d, CXCR3, and CD95 expression in CD4 T cells. For the list of antibodies used for sorting, refer to Supplementary Table 3. After 18-h incubation with PMA/Ionomycin, supernatants were collected and tested by Luminex assay. The Milliplex HTH17MAG-14K (Millipore) was used according to the manufacturer's instructions and signal detected by Flexmap.

The Milliplex HCYTOMAG-60K (Millipore) was used according to the manufacturer's instructions to measure cytokines in the plasma of young and old donors.

**ELISA**. Soluble CD14 and Galectin-9 (R&D Systems), soluble CD163 and IL-21 (eBioscience), IL-26, DKK-1, and SFRP1 (USCN) were measured according to the manufacturer's instructions. Plasma samples of HIV-infected patients and corresponding controls were Triton-inactivated and diluted in the appropriate buffer and assayed in duplicate.

**Autoantibody profiling using the ImmunomeTM protein arrays**. Plasma samples were assayed using the ImmunomeTM protein array (Sengenics Corporation, Singapore)[78]. The array contains quadruplicate spots of 1627 full-length, correctly folded, and fully functional immobilized self- and cancer proteins. These include cancer antigens (mainly cancer–testis antigens (CTAs)), transcription factors, kinases, signaling proteins, and others. Raw data were processed and normalized using a robust customized pipeline[78].

**In vivo transfer into NSG mice of induced $T_{SCM}$ CD4 cells**. Seven days before the transfer, CD4 naive T cells were FACS sorted from aged ($n = 2$) and young ($n = 2$) healthy control's PBMC as CD45RO⁻CCR7⁺CD27⁺CD95⁻ and activated with aCD3/aCD28 magnetic beads (Invitrogen) (1:2 bead:cells ratio) in the presence of IL-7 and IL-15 (10 ng/ml each, Peprotech). Purity of sorted naive CD4⁺ T cells was >97% (not shown). At day 0, magnetic beads were detached and in vitro generated CD4 $T_{SCM}$-enriched cells ($8 \times 10^6$/mouse) were co-transfer with ($5–10 \times 10^6$) CD4-depleted autologous PBMCs obtained by negative magnetic separation with MACS beads (Miltenyi). Mice were weighed every week. Three (day 21; Exp#1) or 4 (day 28; Exp#2) weeks after the transfer, mice were killed, spleens and lungs were collected, weighed, dissociated into single-cell suspension, stained with fluorochrome-conjugated antibodies and analyzed by flow cytometry (LSR Fortessa, BD).

**In vitro induction of $T_{SCM}$ CD4 cells**. CD4 naive T cells were FACS sorted from aged ($n = 15$) and young ($n = 25$) healthy donor's PBMC as CD45RO⁻CCR7⁺CD27⁺CD95⁻ and activated with aCD3/aCD28 magnetic beads (Invitrogen) (1:2 bead:cells ratio) in the presence of DMSO or TWS119 (5 and 10 μM). At day 7, magnetic beads were detached, and in vitro-induced CD4 $T_{SCM}$ were studied for their phenotype and gene expression.

**Quantitative real-time PCR**. Sorted CD4 T-cell subsets were immediately lysed. RNA extraction was performed using an RNeasy Plus Micro kit (Qiagen) and reverse transcribed into cDNA using the SuperScript First Strand kit (Invitrogen). cDNA was analyzed by real-time PCR with the KAPA SYBR qPCR Master Mix kit (KAPA Biosystems) or TAQMAN. The following primers were provided by Qiagen: *BATF* (QT00078449), *IRF4* (QT00065716), *HDAC1* (QT00015239), *PCNA* (QT00024633), or by TAQMAN: *LEF1* (Hs01547250_m1), *TCF7* (Hs01556515_m1), and *Notch1* (Hs01062014_m1).

**nCounter Human Inflammation v2**. Direct mRNA expression levels of the samples were measured using the NanoString nCounter gene expression system. In all, 18,125–20,714 sorted CD4 T-cell subsets in 5 μL of RLT buffer from Qiagen RNeasy Mini kit (Qiagen, Hilden, Germany) were hybridized with probes from the nCounter Human Inflammation v2 panel (Nanostring, Seattle, USA) at 65 °C for 16–19 h according to the nCounter™ Gene Expression Assay Manual. Excess probes were washed away using a two-step magnetic bead-based purification on the nCounter™ Prep Station (GEN1). The nCounter™ Digital Analyzer (GEN1) was used to count individual fluorescent barcodes and quantify target molecules present in each sample. For each assay, a high-density scan (600 fields of view) was performed.

**RNA-seq**. The total RNA was extracted following the double-extraction protocol: RNA isolation by acid guanidinium thiocyanate–phenol–chloroform extraction (TRIzol, Thermo Fisher Scientific, Waltham, MA, USA) followed by a Qiagen RNeasy Micro clean-up procedure (Qiagen, Hilden, Germany). All human RNAs were analyzed on the Agilent Bioanalyzer for quality assessment with RNA integrity number (RIN) range from 6.2 to 9.6 and median RIN 8.9 (Agilent, Santa Clara, CA, USA). cDNA libraries were prepared using 1 ng of the total RNA and 0.5 μl of a 1:50,000 dilution of ERCC RNA Spike in Controls (Ambion Thermo Fisher Scientific, Waltham, MA, USA) using SMARTSeq v2 protocol[79], except for the following modifications: (1) use of 20 μM TSO, (2) use of 250 pg of cDNA with 1/5 reaction of Illumina Nextera XT kit (Illumina, San Diego, CA, USA). The length distribution of the cDNA libraries was monitored using DNA High Sensitivity Reagent Kit on the Perkin Elmer Labchip (Perkin Elmer, Waltham, MA, USA). All samples were subjected to an indexed PE sequencing run of $2 \times 51$ cycles on an Illumina HiSeq 2000 (16 samples/lane). The paired-end reads were mapped to Human GRCh38 reference genome using the STAR alignment tool. The number of reads mapped to each gene was counted using feature Counts (part of Subread package) and GENCODE gene annotation version V25.

**Single cells RNA-seq**. Single-cell cDNA libraries were using the SMARTSeq v2 protocol[79] with the following modifications: (1) 1 mg/ml BSA lysis buffer (Ambion Thermo Fisher Scientific, Waltham, MA, USA); (2) use of 250 pg of cDNA with 1/5 reaction of Illumina Nextera XT kit (Illumina, San Diego, CA, USA). The length distribution of the cDNA libraries was monitored using a DNA High Sensitivity Reagent Kit on the Perkin Elmer Labchip (Perkin Elmer, Waltham, MA, USA). All samples were subjected to an indexed paired-end sequencing run of $2 \times 51$ cycles on an Illumina HiSeq 2000 system (Illumina, San Diego, CA, USA) (192 samples/lane). Pair-endraw reads were aligned to human reference genome using RSEM version 1.3.0. Human reference genome version 25 released by Gencode was used (https://www.gencodegenes.org/human/release_25.html). Transcript Per Million read(TPM) values were calculated using RSEM version 1.3.0 and used for downstream analysis.

**Confocal microscopy**. *Mitochondria contents*: Sorted cells were stained with 100 nm MitoTracker Green (Life Technologies) and 1 μg/ml Hoechst 33258 for 1 h in a humidified incubator. Prior to imaging, cells were washed and resuspended in

R-10 before plating in an eight-well glass-bottom μ-plate (ibidi, Germany). Images were taken using the FV-1000 confocal microscope system (Olympus) under controlled temperature conditions using a 60× oil objective. Mitochondrial staining was analyzed using Imaris (Bitplane, Switzerland).

*Cdc42 polarization*: Frozen sorted naive CD4 T cells subsets were thawed and activated during 2 h at 37 °C with reversible anti-CD3/CD28 Streptamers (IBA Lifesciences, Germany). T cells were seeded on fibronectin-coated glass coverslips in PBS + 10% FBS. After 2 h of incubation at 37 °C (5% $CO_2$) in RPMI-20% FCS with 1% antibiotics, cells were fixed with 4% PFA. After fixation cells were gently washed with PBS, permeabilized with 0.2% Triton X-100 (Sigma) in PBS for 20 min, and blocked with 10% donkey serum (Sigma) for 30 min. Primary and secondary antibody incubations were performed overnight at 4 °C and for 1 h at room temperature, respectively. Coverslips were mounted with ProLong Gold Antifade Reagent with or without DAPI (Invitrogen, Molecular Probes). The cells were coimmunostained with an anti-alpha tubulin antibody (Abcam, rat monoclonal ab6160) detected with an anti-rat AMCA-conjugated secondary antibody or an anti-rat DyLight488-conjugated antibody (Jackson ImmunoResearch), an anti-Cdc42 antibody (Millipore, rabbit polyclonal). Samples were imaged with an AxioObserver Z1 microscope (Zeiss) equipped with a 63 × PH objective. Images were analyzed with AxioVision 4.6 software. Alternatively, samples were analyzed with an LSM710 confocal microscope (Zeiss) equipped with a 63× objective. Primary raw data were imported into the Velocity Software package (Version 6.0, Perkin Elmer) for further processing and conversion into 3D images. As for polarity scoring, the localization of each single-stained protein was considered polarized when a clear asymmetric distribution was visible by drawing a line across the middle of the cell. A total of 50–100 naive CD4 T cells were singularly analyzed per sample. The data are plotted as percentage of the total number of cells scored per sample.

*Scanning electron microscopy*: For imaging by scanning electron microscopy, sorted cells were fixed in 2.5% glutaraldehyde in 0.1 M phosphate buffer for 1 h (pH 7.4) at room temperature, treated post fixation with 1% osmium tetroxide (Ted Pella Inc) at room temperature for 1 h, and then dehydrated through a graded ethanol series from 25 to 100% and critical point dried using a CPD 030 critical point dryer (Bal-Tec AG, Liechtenstein). The cell surfaces on which the cells were grown coated with, and the adhesive surface was coated with 15 nm of gold by sputter coating using a SCD005 high-vacuum sputter coater (Bal-Tec AG). The coated samples were examined with a field emission JSM-6701F Scanning Electron Microscope (JEOL Ltd., USA) at an acceleration voltage of 8 kV using the in-lens secondary electron detector.

**Data analysis**. Flow cytometry data were analyzed using FlowJo (Treestar) and FACSDiva (BD Biosciences). Samples were compared using GraphPad Prism software (v.8.0c). Unbiased t-SNE analysis of flow cytometry data: unbiased representations of multiparameter flow cytometry data were obtained using the t-distributed stochastic neighbor-embedding (t-SNE) algorithm. t-SNE is a nonlinear dimensionality reduction method that optimally locates cells with similar expression levels near to each other and cells with dissimilar expression levels further apart. t-SNE and UMAP analysis and were performed using FlowJo, custom R scripts, and Cytofkit software. Gene expression: Metascape, STEM (Short Time-series Expression Matrix) analysis were performed on the differentially expressed genes between naive T-cell subsets and during aging. For differentially expressed gene (DEG) analysis of RNA-seq data, comparisons between each condition of young and old donors were performed using edgeR, and DEGs were chosen with Benjamini–Hochberg adjusted $p$-values < 0.05. RPKM (reads per kilobase of transcript per million mapped reads) values were computed using edgeR R package version 3.3.2 by filtering the genes that have zero total count and by normalizing the count with TMM algorithm. The averages of RPKM values by condition have been used for STEM analysis. TCR repertoire analysis was performed with MiXCR. Similar amount of mRNA from sorted naive T-cell subsets and identical sequence reads were analyzed to normalize the results. scRNAseq: Young and older donors have been analyzed separately. For both, the donor effect was erased by removing some genes differentially expressed between donors, 183 genes for young donors and 216 genes for old donors, in order to match the density plots of each donor onto the first two PCs. These removed genes were not involved in the population differentiation. We filtered out low-quality cells from our data set based on a threshold for the number of genes detected (a minimum of 200 unique genes per cell). All genes that were not detected in at least 0.4% of all our single cells were discarded, leaving 22,116 genes for young donors and 21,786 for old donors for all further analysis. Normalized data were log-transformed—log(expression+1)—for all downstream analyses. t-SNE and clustering of cells were performed using Seurat R package (https://github.com/satijalab/seurat). FACS data were transformed with a logicle transformation. To summarize the expression of FACS data into each cluster found with the single-cell RNA-sequencing data, a "cluster score" was calculated as the average protein expression in each cluster and row scaled before generating a heatmap using the pheatmap R package. In order to determine the enrichment score for each cluster of $T_{SCM}$ CD4 cells population regarding the Wnt signaling pathway (GO:0016055) and the inflammatory response pathway (GO:0006954), we performed a gene set enrichment analysis (GSEA v3.0). The pathways were retrieved from AMIGO2 website. Monocle analysis was performed using the monocle R package version 2.10[80].

**Statistical analysis**. Groups of young and elderly donors were analyzed by Mann–Whitney U test to compare values. The Wilcoxon matched-pairs signed-rank test was used for paired testing of median values of different subsets from the same donor. Analysis with $p < 0.05$ (*), $p < 0.01$ (**), $p < 0.001$ (***), and $p < 0.0001$ (****) were considered significantly different between the groups.

**Reporting summary**. Further information on research design is available in the Nature Research Reporting Summary linked to this article.

## Data availability
The authors declare that the data supporting the findings of this study are available within the paper (and its supplementary information files). The source data underlying Figs. 2a–g, 4b–c, 5c–g, and 6a–f and Supplementary Figs. 1, 2, 3, 5, 6, and 7 are provided as a Source Data file. The RNA-Seq/scRNAseq have been deposited in in NCBI's Gene Expression Omnibus and are accessible through GEO Series accession number GSE143215 (including sub-series GSE143214 and GSE143213). The data that support the other findings of this study are available from the corresponding author upon reasonable request.

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

## Acknowledgements

We thank Ivy Low (Flow Cytometry Platform), Esther Mok (Multiplex Analysis Platform), Alicia Seok Wei Tay, and Foo Shihui (Genomic Platform) from SIgN

Immunomonitoring Platform (BMRC IAF 311006 grant and BMRC transition funds #H16/99/b0/011). We thank BD Biosciences and BioMarketing Services (BMS) for the generous loan of a BD FACS celesta at University of Malaya. We would like to acknowledge Dr. Robert Balderas (BD Biosciences) for his input in antibody panel design and high-dimensional flow cytometry experiments. We thank Dr. Etienne Becht and Dr. Evan W. Newell for the customized t-SNE and UMAP scripts for flow cytometry unbiased analysis. We thank Dr. Petronela Ancuta (University of Montreal) and Dr. Laurent Reina (SIgN) for the critical reading of the paper. The study is supported by a research grant from the Agency for Science, Technology and Research (No. 10-036), by the Singapore Immunology Network and by a Starting Grant from the European Research Council (ERC-StG-2014 PERSYST 640511 to E.L.). A.L. is a scholar of International Society for Advancement of Cytometry (ISAC). R.R. and A.K. are funded by the High Impact Research/Ministry of Higher Education Research Grant, Malaysia (HIR/MOHE; H-20001-E000001) and the RU grant (UMRG RP029-14HTM). E.R-M was supported by Consejería de Salud y Bienestar Social of Junta de Andalucía through the Nicolás Monardes Program (C-0032/17) and Fondo de Investigación Sanitaria, Instituto de Salud Carlos III, Fondos Europeos para el Desarrollo Regional, FEDER, grants PI16/00684, PI19/01127, RETICS, Red de Investigación en SIDA (RD16/0025/0020).

## Author contributions

Conceptualization, H.K. and A.L.; experimental work, H.K., S.W.T., C.T., M.S., B.M., K.P., V.Z. and J.L.; samples acquisition, R.R., A.K., N.M.G., F.G., E.L. and T.P.N.; data analysis, H.K., M.C.L., M.C., W.H., B.L., A.A., M.C.F., J.M.C., B.M., E.R.M., T.F., H.G. and E.L.; writing—original the draft, H.K.; writing—review and editing, A.L. and G.W.; supervision, H.K. and A.L.; funding acquisition, A.L., T.P.N. and R.R.

## Competing interests

The authors declare no competing interests.
