## [Peer Review File · Nature Communications]

Reviewers' comments:

Reviewer #1, expert in T cell memory (Remarks to the Author):

Kared, et.al. characterize stem cell memory CD4+ T cells through flow cytometry and single cell RNA sequencing samples from young and chronologically aged donors and also some data encompassing inflammation-induced aging (HIV patients before and after HAART therapy). They conclude: reduced frequencies of cells with a stem cell memory phenotype with age and inflammation; deficiencies in the Wnt signaling pathway in the aged (reduced SLAMF6 expression and SFRP1; increased TCF-1 and DKK1); reduced proliferation and cytokine production by stem cell memory; and that recent thymic emigrants are the most likely precursors of CD4 stem cell memory. The manuscript contains a large amount of data with a comprehensive assessment of CD4+ T cell populations and the Wnt signaling pathway. It sometimes lacks clarity, and I have indicated a few places below for modification. The data certainly support a role for RTE as precursors to TSCM and this is an important finding, but I think the data is overstated as it stands and I would like to see the authors strengthen this point further.

The markers used to define the various T cell populations needs to be more clearly described early in the manuscript. A table in Figure 1 listing all of the subsets and the markers used for gating would help the reader OR perhaps switching Supplemental Figure 1 with Figure 1A would work too.

Virtual Memory Cells I believe are referred to here as TMNP(?) but are gated on CD49dhi cells. Virtual memory cells are CD49dlo. (Quinn, et.al., Cell Reports, 2018).

What was the phenotype/purity of CD4 T cells transferred into NSG mice? (Figure 3D) It's important to show that these cells are of the same TSCM phenotype defined earlier in the paper. Phenotypic changes related to the in vitro expansion of this population are referred to but not shown.

The description of Figure 5 is overstated as shown. I see the positive correlation between CD31+PTK7+ (RTE) and iTSCM, but there are also iTSCM coming from the CD31- (Figure 5F) so precursors are present in both populations. Can the data be added to Figure 5D for the CD31- starting population and CD31+PTK7- population?
Also in this figure, are 5E and 5G from young or old donors? What does 5F look like with the 10 μ M stimulation?

Figure 3E/F: What tissue is this data from? Spleen?

Line 312: 'TSCM from young donors preferentially developed into Cluster III cells.' It looks like in the figure the dominant population that forms is Cluster II.

Lines 194-195: hanging sentence. Needs editing.

Figure 3C, S2: color scale is missing

Line 377-378: Citation should be added for generation of CD4 TSCM using TWS119 (Restifo).

Reviewer #2, expert in human T cell profiling (Remarks to the Author):

This manuscript describes experiments dissecting the T cell heterogeneity in humans at various age

("young" vs "old") and in HIV infected patients. With diverse techniques the authors describe their analyses focused on CD4 stem-cell memory T cells (TSCM). It is argued that the Wnt signaling pathway plays a major role in the regulation of this T cell subset, and that the TSCM heterogeneity is due to differential Wnt signaling. Moreover, similar mechanisms may also be involved in age and inflammation associated TSCM decline.

The study provides a wealth of information, particularly phenotype/profiling data of T cell subsets. It also has the merit of a strong focus on memory subsets of CD4 T cells, in contrast to the majority of previous studies that are primarily investigating CD8 memory T cells and their roles in immune responses.

A problem that arises on several occasions is that the authors do not sufficiently distinguish between mechanistic proof vs. associations, and consequently over-interpret the data. Too often the wording is chosen as if the authors can know the underlying mechanisms despite that this is not the case. For example, the authors write "the heterogeneity of TSCM due to differential engagement of Wnt signaling". Furthermore, they write "...key molecules involved in the specific signature of individual TSCM cluster" (why "key"?). Moreover, they describe that they analyzed gene modulation. However, the data do not show functional dynamics. The manuscript should be written with more adequate phrasing such that it becomes clear whenever the authors discover associations but without showing decisive functional involvement, i.e. mechanisms of, and implications in specific functions and/or in vivo importance.

It remains unclear what is meant by old age and young age. In the experimental procedures section, it is mentioned that 78 fresh PBMC samples from individuals from a cohort of old individuals were studied, in parallel to 23 fresh PBMC samples from young individuals. However, it is not known how many individuals were analyzed in total, and there is neither an overview nor a reference of the cohort. Curiously for a study focusing on age, the actual age of the "old" and "young" individuals studied is mostly unclear, and further information about these individuals remains scarce. The numbers of individuals studied in each part are not always indicated. Apparently they were selected differently for the different parts of the study, but the criteria for selection are missing!

Besides occasional remarks, it remains unknown whether the analyses were done with fresh or frozen PBMC. For the latter, were the cells frozen in liquid nitrogen, were they shipped and if yes how?

The abstract mentions defects in TSCM in elderly, and reports about an increase in DKK1. However, in the results section the authors write: "The increased expression of DKK1 by TSCM in elderly donors, was therefore not a plausible explanation for the loss of Wnt/ β -Catenin signaling integrity" (lines 370-372). It is difficult to justify putting DKK1 forward so prominently in the abstract.

Longitudinal studies would be preferable to the side-by-side investigation of young vs. old individuals that certainly differ not only for age but also for many other and potentially confounding parameters. This should be discussed, and whenever longitudinal data and corresponding references are available they should be put forward and cited.

What is the evidence that elderly healthy individuals have "excessive inflammation" (line 254)?

The term "canonical memory T cell precursors" is proposed for TSCM, HPCs, TRTE and naïve T cells (lines 440-441). It remains unclear what the authors mean with this term and why they think it should be used. What is meant by HPCs?

Are the data for TCR diversity corrected/normalized for the differing numbers of cells analyzed?

Further points:

The English is not always easy to understand.

For example, the grammar of the sentence starting in line 145 may be improved.

The same is suggested for "leading promoting diseases" (line 182),

and "The explanation for the age-related disturbed Wnt signaling in TSCM was further investigated (lines 246/247).

There are many more instances.

Please specify what is meant by "...and showed reversibility." (lines 194/195)? (and correct the typo in "reversibility")

Please correct "...with the limitation that host re young animals...", and "impedes" (line 481)

Reviewer #3, expert in clinical and computational immunology (Remarks to the Author):

The manuscript "Immunological History Governs Human Stem Cell Memory CD4 Heterogeneity Via The Wnt Signaling Pathway" measures phenotypic and functional variation of stem cell memory CD4 T cells from young and aged individuals, pinpoints Wnt signaling as affected by age, and identifies recent thymic emigrants as a putative source of stem cell memory populations. Investigating the origins of this population is helpful to understanding immune changes with age, inflammation, and chronic infection. In particular, while a number of studies have investigated CD8 T cells of this phenotype, studies on CD4 stem cell memory have been very limited to date. This study contains a number of intriguing findings. However, there are several major concerns that significantly limit the enthusiasm of this reviewer. These are listed below, along with a few minor suggestions:

Major concerns:

- 1) There are critical details of the cell origin including age distribution and history of chronic/latent infection that are not included in the current manuscript. Noting that these subjects were <35 or >65 years of age and part of a larger cohort is not adequate to understand the data. For example, history of CMV or EBV infection could shape the T cell repertoire and differentiation state including Tscm and Tscm precursors. Without further information on these and other clinical characteristics of the "young" and "aged" cohort, it is impossible to adequately interpret the data presented.
- 2) The authors argue that RTE are the precursors of CD4 TSCM. However, data presented indicate that either RTE or non-RTE can differentiate into TSCM. It would be important to determine if CD31negative naïve cells are an intermediate cell type in this differentiation.
- 3) The authors use a system of in vitro expansion followed by adoptive transfer into NSG mouse model as support for in vivo TSCM differentiation. Expansion in vivo confounds any potential conclusions about what is really happening in vivo.
- 4) The data on Wnt/ β -catenin signaling are largely correlative, and the conclusions about impacts on Wnt signaling are not fully justified. As an example, data are presented to suggest that the DKK1 is elevated with aging and this inhibits Wnt signaling. However, the final two sentences of this section of Results (Regulation of human TSCM CD4 cells homeostasis: the Wnt/DKK1 axis) indicate that DKK1 changes do NOT explain changes in Wnt signaling, but no alternative is proposed.

Minor concerns:

- 1) Three populations of TSCM are identified based on gene expression and described as if they are functionally distinct, but the data on differences in Wnt signaling are limited to gene expression and not functional proof.

- 2) TSCM are described in the manuscript as “naïve” and “antigen-inexperienced,” but they are clearly distinct from naïve cells, and have antigen experience.
- 3) Under “multiplex analytes screening” in the Experimental Procedures, there is a reference to Supplementary Table 2, which does not appear to be present.
- 4) 7 day Wnt stimulation likely has very different impacts on RTE differentiation from in vivo homeostatic and antigen-driven stimulation. The data in Figures 5B, S5A, and S5B show a number of phenotypic differences between in vitro generated and ex vivo TSCM. Thus, there is concern about the validity of this in vitro approach to study TSCM.
- 5) There are several ideas not integrated into the story as a whole. Specific examples include the following. HIV is used as a model of chronic infection, but the abstract and introduction do not mention HIV at all. The autoantibody analysis is clearly of antibodies to genes in the Wnt pathway, but it is never discussed why antibodies to these proteins would be expected.

Reviewer 1

We are glad that this expert on T-cell memory has appreciated our work. We have taken into consideration her/his concerns to restrain our enthusiasm and avoid overstating our claims. The modifications are highlighted through the revised manuscript.

We respond accordingly to the specific points:

1. *The markers used to define the various T cell populations needs to be more clearly described early in the manuscript. A table in Figure 1 listing all of the subsets and the markers used for gating would help the reader OR perhaps switching Supplemental Figure 1 with Figure 1A would work too.*

The markers used to define the various T cell populations are now listed in table 1.

Name	Abbreviation	Phenotype	notes
Recent Thymic Emigrants	T _{RTe}	CCR7 ⁺ CD45RO ⁻ CD27 ⁺ CD28 ⁺ CD62L ⁺ CD122 ⁻ CD95 ⁻ CD31 ^{high} PTK7 ⁺	CD62L only used in freshly isolated cells
Naïve	T _{NAIVE}	CCR7 ⁺ CD45RO ⁻ CD27 ⁺ CD28 ⁺ CD62L ⁺ CD122 ⁻ CD95 ⁻ CD31 ^{dim}	CD62L only used in freshly isolated cells
Memory T cells with a Naïve Phenotype	T _{MNP}	CCR7 ⁺ CD45RO ⁻ CD27 ⁺ CD28 ⁺ CD122 ⁻ CD95 ⁻ CD31 ⁻ CD49d ^{high} CXCR3 ⁺	
Stem Cell Memory	T _{SCM}	CCR7 ⁺ CD45RO ⁻ CD27 ⁺ CD62L ⁺ CD122 ⁻ CD95 ⁻	CD62L only used in freshly isolated cells
Virtual Memory	T _{VM}	CCR7 ⁺ CD45RO ⁻ CD27 ⁺ CD5 ⁺ CD95 ⁻	Better identification with CD122 ⁺ CD127 ⁺ PanKIR ⁺ NKG2A ⁺ Eomes ⁺
Central Memory	T _{CM}	CCR7 ⁺ CD45RO ⁻ CD27 ⁺ CD28 ⁺ CD95 ⁻	
Transitional Memory	T _{TM}	CCR7 ⁺ CD45RO ^{int} CD27 ^{int} CD28 ⁺ CD95 ⁻	TM1/2 could be distinguish based on CD45RO and CD27 expression
Effector Memory	T _{EM}	CCR7 ⁻ CD45RO ⁺ CD27 ⁻ CD28 ⁺ CD95 ⁻	
Terminal Effector Memory RA	T _{EMRA}	CCR7 ⁻ CD45RO ⁺ CD45RA ⁺ CD27 ⁻ CD28 ⁺ CD95 ⁻	

2. *Virtual Memory Cells I believe are referred to here as TMNP(?) but are gated on CD49dhi cells. Virtual memory cells are CD49dlo. (Quinn, et.al., Cell Reports, 2018).*

We thank the reviewer for this question related to Virtual Memory CD4 T cells. We regret being unclear that this population increases during aging (Quinn et al, 2018; White, 2016, Jacomet, 2015). These studies have focused on T_{VM} CD8 cells in mice and have described this population as CD44^{high}CD49d^{low}. However, T_{VM} have also been characterized as CD5^{-/low}CD27⁺CD122⁺Eomes⁺PAN-KIR⁺NKG2A⁺ in Human T cells (Quinn et al, 2018; White, 2016, Jacomet, 2015); the use of CD27 and CD5 enabled us to distinguish T_{VM} CD4 cells in our cohort from T_{MNP} and T_{SCM}. Further

investigation is needed to delineate these different subsets. However, it is beyond the scope of this study.

3. **What was the phenotype/purity of CD4 T cells transferred into NSG mice? (Figure 3D) It's important to show that these cells are of the same TSCM phenotype defined earlier in the paper. Phenotypic changes related to the *in vitro* expansion of this population are referred to but not shown.**

The phenotype/purity of CD4 T cells that was transferred into NSG is indeed important for determining the therapeutic applications of expanded cells. We have included the overlaid phenotype of gated naïve CD4 T cells at day 0 and induced T_{SCM} at day 7 (as per below) (Fig.S3C). This should help clarify the phenotype and putative effects of *in vitro* expansion.

4. **The description of Figure 5 is overstated as shown. I see the positive correlation between CD31⁺PTK7⁺ (RTE) and iTSCM, but there are also iTSCM coming from the CD31⁻ (Figure 5F) so precursors are present in both populations. Can the data be added to Figure 5D for the CD31⁻ starting population and CD31⁺PTK7⁻ population?**

Also in this figure, are 5E and 5G from young or old donors? What does 5F look like with the 10 μ M stimulation?

We apologize for the overstatement in figure 5. It is correct that iT_{SCM} could also be induced *in vitro* from CD31⁻ naïve CD4 T cells through Wnt signaling activation. We have edited the text and included here the other naïve populations (CD31⁻PTK7⁻ and CD31⁺PTK7⁻). We did not observe any correlation between naïve CD4 T cells non-expressing PTK7 at Day 0 and the induction of T_{SCM} at day 7.

We have included in the figures 5E-G, young and old donors (identified in figure 5F by black and grey symbols respectively). As presented in figure 5G, the highest dose of TWS119 (10uM) did not change the phenotype of iT_{SCM}. We believe the modifications provided clarify on the origin of T_{SCM} still highlighting the higher ability of T_{RTE} to give rise to T_{SCM}.

5. *Figure 3E/F: What tissue is this data from? Spleen?*

The tissue origin is now stated in the figure legend. These cells were collected from lung and spleen and analyzed simultaneously.

6. *Line 312: 'TSCM from young donors preferentially developed into Cluster III cells.' It looks like in the figure the dominant population that forms is Cluster II*

We regret the typo error in line 312. We agree that the main population associated with Young iT_{SCM} is cluster II. Our point was to highlight that cluster III was more developed in animals receiving iT_{SCM} from young than old animals. We have removed this set of data to improve the readiness of the manuscript.

7. *Lines 194-195: hanging sentence. Needs editing. Figure 3C, S2: color scale is missing Line 377-378: Citation should be added for generation of CD4 TSCM using TWS119 (Restifo).*

The text, citations and figures have been corrected accordingly. We greatly appreciated the reviewer's time and elaborated feedback, which we believe significantly improved clarify and readiness of the paper.

Reviewer 2.

We thank the reviewer for his positive feedback. We have taken into consideration all his comments to rephrase the text in order to temper our claims, and stick to the facts. We edited the text to more adequately distinguish between causation and association. This was also a comment from Reviewer 1.

1. *A problem that arises on several occasions is that the authors do not sufficiently distinguish between mechanistic proof vs. associations, and consequently over-interpret the data. Too often the wording is chosen as if the authors can know the underlying mechanisms despite that this is not the case. For example, the authors write “the heterogeneity of TSCM due to differential engagement of Wnt signaling”. Furthermore, they write “...key molecules involved in the specific signature of individual TSCM cluster” (why “key”?). Moreover, they describe that they analyzed gene modulation. However, the data do not show functional dynamics. The manuscript should be written with more adequate phrasing such that it becomes clear whenever the authors discover associations but without showing decisive functional involvement, i.e. mechanisms of, and implications in specific functions and/or in vivo importance.*

We tried to avoid the excessive use of extensive figures and data, so we excluded changes in gene modulation between naïve CD4 T cells subsets. The *in silico* STEM analysis describes the functional dynamics of genes (Fig.S5C-D). This model reflects the highest probability of gene modulation but remains hypothetical. We evaluated the putative ontogeny (or itinerary) and relationship between our cells of interest. IPA analyzed the dynamic and the nature of genes involved in this progression. The most significant association is described below. The main genes involved differed during aging and referred to different metabolic pathways.

Top Canonical Pathways

Name	Young Donors	p-value	Cluster 11	Name	Old Donors	p-value
Apoptosis Signaling	3.62E-05			Sirtuin Signaling Pathway	9.22E-05	
Death Receptor Signaling	1.32E-04			Oxidative Phosphorylation	2.65E-04	
CDP-diacylglycerol Biosynthesis I	3.51E-04			Remodeling of Epithelial Adherens Junctions	5.80E-04	
Phosphatidylglycerol Biosynthesis II (Non-plastidic)	5.85E-04			RhoA Signaling	9.69E-04	
Triacylglycerol Biosynthesis	7.28E-04			Unfolded protein response	1.23E-03	
Name	Young Donors	p-value	Cluster 10	Name	Old Donors	p-value
Antiproliferative Role of TOB in T Cell Signaling	4.08E-02			EIF2 Signaling	3.37E-03	
Circadian Rhythm Signaling	5.85E-02			Regulation of eIF4 and p70S6K Signaling	2.03E-02	
L-carnitine Biosynthesis	5.85E-02			Systemic Lupus Erythematosus Signaling	2.30E-02	
Taurine Biosynthesis	5.85E-02			Hereditary Breast Cancer Signaling	3.10E-02	
L-glutamine Biosynthesis II (tRNA-dependent)	5.85E-02					
Name	Young Donors	p-value	Cluster 12	Name	Old Donors	p-value
EIF2 Signaling	8.9 % 17/191	1.26E-04		mTOR Signaling	3.18E-03	
Regulation of eIF4 and p70S6K Signaling		4.80E-04		Mechanisms of Viral Exit from Host Cells	4.55E-03	
tRNA Charging		9.97E-04		EIF2 Signaling	6.04E-03	
UDP-N-acetyl-D-glucosamine Biosynthesis II		1.39E-02		Regulation of eIF4 and p70S6K Signaling	1.05E-02	
Valine Degradation I		1.82E-02		Macropinocytosis Signaling	2.29E-02	
Name	Young Donors	p-value	Cluster 8	Name	Old Donors	p-value
Regulation of Actin-based Motility by Rho		1.30E-05		Actin Cytoskeleton Signaling	1.65E-03	
TCA Cycle II (Eukaryotic)		1.31E-05		TNFR1 Signaling	1.71E-03	
Aspartate Degradation II		1.79E-04		Sumoylation Pathway 9.2 % 8/87	3.05E-03	
Protein Ubiquitination Pathway		2.03E-04		Regulation of Actin-based Motility by Rho	3.16E-03	
RhoA Signaling		2.50E-04		Glutathione Redox Reactions II	4.55E-03	
Name	Young Donors	p-value	Cluster 5	Name	Old Donors	p-value
Systemic Lupus Erythematosus Signaling		5.05E-04		Tetrapyrrole Biosynthesis II	3.77E-03	
EIF2 Signaling		8.72E-04		Glycogen Biosynthesis II (from UDP-D-Glucose)	6.18E-03	
Serine Biosynthesis		4.41E-03		DNA damage-induced 14-3-3 Signaling	1.01E-02	
Regulation of eIF4 and p70S6K Signaling		5.26E-03		Estrogen Receptor Signaling	1.13E-02	
Superpathway of Serine and Glycine Biosynthesis I		1.06E-02		Heme Biosynthesis II	1.65E-02	

2. It remains unclear what is meant by old age and young age. In the experimental procedures section, it is mentioned that 78 fresh PBMC samples from individuals from a cohort of old individuals were studied, in parallel to 23 fresh PBMC samples from young individuals. However, it is not known how many individuals were analyzed in total, and there is neither an overview nor a reference of the cohort. Curiously for a study focusing on age, the actual age of the “old” and “young” individuals studied is mostly unclear, and further information about these individuals remains scarce. The numbers of individuals studied in each part are not always indicated. Apparently they were selected differently for the different parts of the study, but the criteria for selection are missing!

We thank the reviewer for this point. We have detailed the number of subjects for every experiment. The bio-samples from older adults used in this

study are part of a larger cohort called SLAS (Singaporean Longitudinal Aging Study), whose characteristics have already published, and we have referenced previous publications on this cohort. In order to give a clearer picture of our subjects, we have added an extra-table (Table 2) documenting age distribution, gender, and history of chronic infection. The prevalence of chronic infection is indeed pretty different between young and older donors and may contribute to the inflammatory signature that was detected in T_{SCM}. We are planning to recruit elderly donors with low or high IgG anti-CMV antibodies to dissect better the role of persistent infection (or patients with/without a better control of chronic CMV infection) on T_{SCM} homeostasis. We would like to analyze if CMV status is associated with preferential gene expression and phenotype of T_{SCM}. We have generated preliminary data in HAART-treated HIV cohort regarding T_{SCM} frequency according to CMV sero-status and age. We observed that T_{SCM} frequency was decreased in young (<35 years old) and middle age (35<age<65 years old) HIV patients with elevated anti-CMV IgG titer (data not shown).

	Age (Median ±SD)	Gender (n Male/ Total)	Serology: % positive donors (Median IgG value, U/mL)								Significance (MW U test)
			H Pylori IgG	EBV EBNA1 IgG	VZV IgG	HSV1 IgG	HSV2 IgG	CHKN IgG	Dengue IgG	CMV IgG	
Young	22 ± 5	31/62	5 % (4)	80 % (54)	87 % (886)	35% (21)	0% (2.9)	9% (5)	18% (1)	38% (8)	
Old	68 ± 9	79/219	30 % (12)	98 % (34)	96% (761)	86% (78)	19% (3.3)	1% (2)	87% (32)	97% (781)	
	IgG concentration		****	****	N.S.	****	***	****	****	****	
	neg	<35	<2.5	<50	<20	<20	<9	<9	<25		
	borderline	35-50	2.5-3	50-100	20-30	20-30	9-11	9-11	25-40		
	pos	>50	>3	>100	>30	>30	>11	>11	>40		
	Units	U/ml	U/ml	mIU/ml	U/ml	U/ml	Standard Units	Panbio Units	PEI-U/mL		

The main criterion of selection was based on the availability of the samples depending on the visit of elderly people at the study site. We were particular in performing functional assays and subset quantification (especially for absolute counts) with freshly isolated blood or PBMCs to avoid any bias due to freezing/thawing conditions. The cells and plasma of the entire cohort were also cryo-preserved in the laboratory in liquid nitrogen and at -80C respectively. The cell viability after thawing was >95% with >80% recovery. We have detailed in the manuscript the origin of the cells for each experiment.

3. *The abstract mentions defects in TSCM in elderly, and reports about an increase in DKK1. However, in the results section the authors write: “The increased expression of DKK1 by TSCM in elderly donors, was therefore not a plausible explanation for the loss of Wnt/β-Catenin signaling integrity” (lines 370-372). It is difficult to justify putting DKK1 forward so prominently in the abstract.*

We thank the reviewer for this comment and we apologize for the confusion. Our main hypothesis is that DKK1 may alter the homeostasis of T_{SCM} CD4 cells. DKK1 concentration was increased in the plasma of elderly donors and negatively correlated with the frequency of T_{SCM} CD4 cells. However, the source of DKK1 was unknown (trans-inhibition through secretion by platelets, regulatory T cells, or others as described elsewhere). So, we investigated whether DKK1 could directly come from T_{SCM} and mediate cis-inhibition. We have measured the level of DKK1 mRNA in T_{SCM} subsets during aging and we did not observe significant modulation in older donors. Therefore, we have excluded the hypothesis that DKK1 auto-regulate

T_{SCM} as cells other than T_{SCM} could also secrete DKK1. We have reformulated the sentence to avoid any further confusion.

4. *Longitudinal studies would be preferable to the side-by-side investigation of young vs. old individuals that certainly differ not only for age but also for many other and potentially confounding parameters. This should be discussed, and whenever longitudinal data and corresponding references are available they should be put forward and cited.*

We agree with the reviewer that a longitudinal or family study will represent the most interesting cohort. Unfortunately, this type of study cannot be performed in the current setting for the purpose of this study. Our cohort is based subject to the successful recall of patients, which enable us to closely monitor clinical and biological parameters, but only a few patients are available at the time of the study. But as proposed by this reviewer, this is planned and is awaiting to collect sufficient donors to test the time evolution of T_{SCM} and associate clinical and biological parameters to understand better the dynamic of T_{SCM} during lifetime.

5. *What is the evidence that elderly healthy individuals have “excessive inflammation” (line 254)?*

Inflamm-aging is a notion that summarizes phenomena, which describe chronic inflammation in elderly donors. In order to evaluate the inflammation in our cohort, we measured several inflammatory markers such as CRP, TNF- α , Neopterin, sCD14, sCD163 and, IL-18 and show that these are elevated in healthy elderly during aging, as shown in Figure 2D. These markers are associated with persistent inflammation during chronic infections, such as those described for HIV.

6. *The term “canonical memory T cell precursors” is proposed for TSCM, HPCs, TRTE and naïve T cells (lines 440-441). It remains unclear what the authors mean with this term and why they think it should be used. What is meant by HPCs?*

We refer to cells in the literature that can self-renewed and be differentiated into more mature cells, including T_{SCM}, HPC (hematopoietic progenitor cells) and T_{RTE}. The stemness of these cells and their potential abilities to self-renew and to differentiate into all mature T cells (or all hematopoietic cells for HPC) by asymmetric division characterizes T_{SCM}.

7. *Are the data for TCR diversity corrected/normalized for the differing numbers of cells analyzed?*

We have completed the methodology section to indicate how the measurement of TCR diversity was normalized based on the similar amount of used mRNA and the identical depth of sequencing reads.

8. *The English is not always easy to understand.*

We have revised the text to simplify our message and correct further grammatical errors. We apologized for the inappropriate formulations.

Reviewer 3.

We thank the reviewer for the feedback about the manuscript.

1. *There are critical details of the cell origin including age distribution and history of chronic/latent infection that are not included in the current manuscript. Noting that these subjects were <35 or >65 years of age and part of a larger cohort is not adequate to understand the data. For example, history of CMV or EBV infection could shape the T cell repertoire and differentiation state including Tscm and Tscm precursors. Without further information on these and other clinical characteristics of the “young” and “aged” cohort, it is impossible to adequately interpret the data presented.*

We agree with the reviewer that the history of chronic viral infections could modulate the differentiation of T cell subsets, the repertoire and, characteristics of T_{SCM} precursors. We have detailed the gender and age of participants in Table 2. We have also detailed any detectable history of persistent infections in our cohort of young and old donors. We could provide additional clinical information if the reviewer requires.

2. *The authors argue that RTE are the precursors of CD4 TSCM. However, data presented indicate that either RTE or non-RTE can differentiate into TSCM. It would be important to determine if CD31negative naïve cells are an intermediate cell type in this differentiation.* We thank the reviewer for this point. Our results suggest that T_{RTE} have better innate capacity to differentiate into T_{SCM}. However, intense stimulation of the Wnt pathway enabled the differentiation of CD31^{neg} naïve T cells into T_{SCM} (data below).

Moreover, we have tested several hypotheses of T_{SCM} ontogeny. We have interchanged the starting and intermediate populations in an

unbiased way. Monocle analysis revealed also that CD31^{neg} and T_{RTE} could differentiate into T_{SCM} (Fig.S5E). The most significant model of differentiation integrates CD31^{neg} population as potential intermediate population but a direct path from T_{RTE} to T_{SCM} might coexist too.

3. *The authors use a system of in vitro expansion followed by adoptive transfer into NSG mouse model as support for in vivo TSCM differentiation. Expansion in vivo confounds any potential conclusions about what is really happening in vivo.*

We understand the concern of the reviewer about the in vivo experiments. However, the objective of this experiment was essentially to evaluate the engraftment and *in vivo* proliferation capacities of T_{SCM} that were derived from aged subjects. This methodology enables us to measure the ability of T_{SCM} to undergo asymmetric division into both mature cells - such as T_{CM} and T_{EM}, - and also those with T_{SCM} characteristics. The paucity of CD4 T_{SCM} in peripheral blood from elderly donors prevented us from directly transferring them into NSG mouse. Here, we have benefited from the expertise of Dr. Lugli, who was among the first one to discover and characterize CD8 T_{SCM} in human. So, our strategy was to reproduce his approach to compare human CD4 and CD8 T_{SCM}.

4. *The data on Wnt/β-catenin signaling are largely correlative, and the conclusions about impacts on Wnt signaling are not fully justified. As an example, data are presented to suggest that the DKK1 is elevated with aging and this inhibits Wnt signaling. However, the final two sentences of this section of Results (Regulation of human TSCM CD4 cells homeostasis: the Wnt/DKK1 axis) indicate that DKK1 changes do NOT explain changes in Wnt signaling, but no alternative is proposed.*

We thank the reviewer and understand the limitation of our human study.

We have reformulated our description of results and conclusions to avoid any overstatement. The final two sentences were indeed unclear and have detailed our response for the reviewer 2. Our main conclusion refers to the trans- rather than cis-regulation of T_{SCM} by extrinsic DKK1.

5. *Three populations of TSCM are identified based on gene expression and described as if they are functionally distinct, but the data on differences in Wnt signaling are limited to gene expression and not functional proof.*

The comment of the reviewer on T_{SCM} is relevant. However, the amount of work and cost of single cell sequencing to identify gene expression and phenotype of T_{SCM} subsets has limited our possibility to functionally validate them. This will be the topic of our next study.

6. *TSCM are described in the manuscript as “naïve” and “antigen-inexperienced,” but they are clearly distinct from naïve cells, and have antigen experience.*

We agree that the formulation could be inappropriate. We tried to avoid repetition but these non-interchangeable terms were misleading. We have corrected this error.

7. Under “multiplex analytes screening” in the Experimental Procedures, there is a reference to Supplementary Table 2, which does not appear to be present.

We have corrected and re-inserted all the tables.

8. 7 day Wnt stimulation likely has very different impacts on RTE differentiation from in vivo homeostatic and antigen-driven stimulation. The data in Figures 5B, S5A, and S5B show a number of phenotypic differences between in vitro generated and ex vivo TSCM. Thus, there is concern about the validity of this in vitro approach to study TSCM.

We thank the reviewer for this point. We did not develop enough the discussion about the limitation of this in vitro T_{SCM} induction. The difference between ex-vivo isolated T_{SCM} and iT_{SCM} have already been documented (Lugli and others...) in term of phenotype and gene expression. We are aware that the comparison of resting (freshly isolated T_{SCM}) and stimulated cells (iT_{SCM}) would necessarily induce phenotypic differences.

9. There are several ideas not integrated into the story as a whole. Specific examples include the following. HIV is used as a model of chronic infection, but the abstract and introduction do not mention HIV at all. The autoantibody analysis is clearly of antibodies to genes in the Wnt pathway, but it is never discussed why antibodies to these proteins would be expected.

We thank the reviewer for this final point. We have attempted to be concise and focused our message on inflammation during aging. We have reformulated the abstract and the introduction to integrating HIV into the story. We have also further developed the rationale for studying autoantibodies.

REVIEWERS' COMMENTS:

Reviewer #1 (Remarks to the Author):

The authors have provided additional data and information that improves clarity and strengthens their claims, and this has addressed the major concerns raised on previous review.

There are still multiple grammar issues which need to be corrected. I have also listed some figure corrections and places within the text that incorrectly refer to figures.

Figures 3C, S2A, S5B: still no color scale for heat maps provided.

Line 313: There is no Figure 3F

Line 331: There is no Figure 3G. I believe this should be Figure 3E.

Reviewer #2 (Remarks to the Author):

I thank the authors for the appropriate revisions and have no further comments.

Reviewer #3 (Remarks to the Author):

The authors have substantially revised the initial manuscript and have appropriately addressed several of the issues that were raised. However, while information regarding chronic/latent infection has been added in Table 2, the manuscript would benefit from an additional 1-2 sentences in the discussion of how chronic infection and its effects on the immune repertoire may limit the conclusions; i.e. how do the authors think the difference in previous CMV exposure (seropositivity) between 38% of their "young" cohort versus 97% of the "old" cohort may affect and limit their conclusions as this virus has been shown to shape the T cell repertoire.

Reviewer #1 (Remarks to the Author):

The authors have provided additional data and information that improves clarity and strengthens their claims, and this has addressed the major concerns raised on previous review.

There are still multiple grammar, which need to be corrected. I have also listed some figure corrections and places within the text that incorrectly refer to figures.

Figures 3C, S2A, S5B: still no color scale for heat maps provided.

Line 313: There is no Figure 3F

Line 331: There is no Figure 3G. I believe this should be Figure 3E.

We thank the reviewer for his enthusiasm about our work. We apologized for the errors. We have revised the grammar and typo mistakes. We are sorry about the missing scale in the pdf document. We have reformatted the figures to avoid this issue. We have verified the references of figures after the re-organization of the manuscript to not reproduce this mistake.

Reviewer #2 (Remarks to the Author):

I thank the authors for the appropriate revisions and have no further comments.

Reviewer #3 (Remarks to the Author):

The authors have substantially revised the initial manuscript and have appropriately addressed several of the issues that were raised. However, while information regarding chronic/latent infection has been added in Table 2, the manuscript would benefit from an additional 1-2 sentences in the discussion of how chronic infection and its effects on the immune repertoire may limit the conclusions; i.e. how do the authors think the difference in previous CMV exposure (seropositivity) between 38% of their "young" cohort versus 97% of the "old" cohort may affect and limit their conclusions as this virus has been shown to shape the T cell repertoire.

We are glad that the reviewer has appreciated our revised manuscript. We agree that the influence of persistent infection on the homeostasis of immune cells during aging is a major topic of our study. We have added two more sentences in the discussion to highlight the restriction of TCR repertoire and the clonal expansion of CMV-specific T cells. We discussed about the burden of chronic infections and inflammation on the nature of CD4 T cell response in elderly donors.